# Molecular and electrophysiological features of GABAergic neurons in the dentate gyrus reveal limited homology with cortical interneurons

Quentin Perrenoud[1], Clémence Leclerc[1,2], Hélène Geoffroy[1], Tania Vitalis[1], Kevin Richetin[2], Claire Rampon[2☉], Thierry Gallopin [1☉]*

**1** Brain Plasticity Unit, CNRS, ESPCI Paris, PSL Research University, Paris, France, **2** Centre de Recherches sur la Cognition Animale (CRCA), Centre de Biologie Intégrative (CBI), Université de Toulouse; CNRS, UPS, France

☉ These authors contributed equally to this work.
* thierry.gallopin@espci.fr

**Data Availability Statement:** The data underlying the results presented in the study are available on Dryad, a public repository. URL: https://doi.org/10.

## Abstract

GABAergic interneurons tend to diversify into similar classes across telencephalic regions. However, it remains unclear whether the electrophysiological and molecular properties commonly used to define these classes are discriminant in the hilus of the dentate gyrus. Here, using patch-clamp combined with single cell RT-PCR, we compare the relevance of commonly used electrophysiological and molecular features for the clustering of GABAergic interneurons sampled from the mouse hilus and primary sensory cortex. While unsupervised clustering groups cortical interneurons into well-established classes, it fails to provide a convincing partition of hilar interneurons. Statistical analysis based on resampling indicates that hilar and cortical GABAergic interneurons share limited homology. While our results do not invalidate the use of classical molecular marker in the hilus, they indicate that classes of hilar interneurons defined by the expression of molecular markers do not exhibit strongly discriminating electrophysiological properties.

## Introduction

The dentate gyrus is a major entry point to the hippocampus, a key region for learning, memory, spatial navigation and processing of emotions. It integrates sensory information originating from the entorhinal cortex and is believed to be crucial for the encoding of contextual representations during memory formation and retrieval [1–4]. The dentate gyrus is also one of the rare brain regions where new neurons are generated throughout adult life [5–7]. Finally, it is one of the most vulnerable brain regions to epilepsy, brain trauma, and ischemia [8–12].

The dentate gyrus consists of three layers. The somas of principal neurons, the so-called glutamatergic granule cells, are packed in the granule cell layer (1). The dendrites of the granule cells extend in the molecular layer (2) where they receive inputs coming from the

5061/dryad.37pvmcvp1 DOI: 10.5061/dryad.37pvmcvp1.

**Funding:** This work was supported by a grant from the France Alzheimer Association (www.francealzheimer.org/). The work was also supported by the Centre National de la Recherche Scientifique (www.cnrs.fr/fr), the University of Toulouse 3 (www.univ-toulouse.fr) and the Ecole Supérieure de Physique et Chimie Industrielle-Paris (ESPCI Paris, www.espci.psl.eu/en). The funders had no role in study design, data collection and analysis, decision to publish, or preparation of the manuscript.

**Competing interests:** The authors have declared that no competing interests exist.

entorhinal cortex and contralateral hippocampus through the perforant and commissural pathways. The hilus (3) sits on the opposite side and contains glutamatergic mossy cells and GABAergic interneurons [13]. Hilar GABAergic interneurons are critical elements of the dentate circuitry organizing the activity of granule cells through lateral and feedback inhibition and gating afferent inputs [14–17]. Hilar GABAergic interneurons display a wide variety of firing patterns, axonal arborizations and molecular contents [13,18–21] and their diversity is still not fully understood [14].

Most GABAergic interneurons populating the telencephalon (i.e. the hippocampus, cortex and striatum) share a common developmental origin [22–25]. Accordingly, these interneurons tend to diversify into a handful of major classes with recognizable morphology, electrophysiology, and molecular expression [26–33]. At the molecular level, these classes are well defined based on the expression of calcium binding proteins such as parvalbumin (PV) and calretinin (CR), and neuropeptides like somatostatin (SOM), Neuropeptide Y (NPY) and cholecystokinin (CCK), and enzymes such as the neuronal NO synthase (NOS1). In the hilus, the expression of PV, SOM, NOS1 and CCK is now widely used to select specific populations of neurons by breeding Cre-recombinase mouse lines [16,34–39]. Transgenic labeling has convincingly shown that PV and SOM associate with specific patterns of projection in the granule cell and molecular layer respectively [16,34,39,40]. However, to date, the co-expression pattern of commonly used markers has not been thoroughly investigated, leaving the possibility that overlapping classes are labeled in different mouse lines. In addition, few studies have attempted a multiparametric classification of hilar interneurons. In these studies, it has historically been difficult to identify meaningful combinations of features, leading some authors to speculate that hilar interneurons form a "continuum" where each cell is unique [20,41]. Therefore, it remains unclear how the GABAergic interneurons in the dentate gyrus differ from those located in other telencephalic regions.

In an attempt to resolve this issue, we are examining whether electrophysiological and molecular features classically used to group GABAergic interneurons in the forebrain, can also appropriately capture the diversity of hilar GABAergic interneurons. First, using transgenic mice expressing the green fluorescent protein (GFP) under the control of the glutamic acid decarboxylase 67k (GAD67) promoter [42], we characterized the distribution of GABAergic interneurons immunoreactive for CR, PV, SOM, NPY and NOS1 throughout the hilus and granular cell layer. Then, we compared the relevance of electrophysiological and molecular features for the classification of GABAergic interneurons sampled from the hilus and primary sensory cortex, using patch clamp combined with single cell RT-PCR, a highly sensitive method previously employed by our group and others to classify interneurons in the cortex and hippocampus [43–48]. Finally, using resampling based statistical analysis, we substantiated the conclusion that hilar and cortical GABAergic interneurons only share a limited phenotypic homology. Our findings indicate that markers such as PV, NOS1 and CCK are associated with a unique combination of electrophysiological and molecular features in hilar GABAergic interneurons.

## Materials and methods

### Immunohistochemistry

Eight P50 GAD67-GFP knock-in (Δneo) transgenic mice [42] maintained on a C57BL/6 genetic background were used for this study. Mice were kindly provided by Dr Y. Yanagawa (National Institute for Physiological Science, Okazaki, Japan). Mice were deeply anaesthetized with an intraperitoneal injection of Pentobarbital (150 mg/kg body weight) and perfused transcardially with 4% paraformaldehyde in 0,1M saline phosphate buffer (PBS), pH 7.4.

Brains were dissected out, embedded in 3% agarose diluted in PBS and cut coronally on a vibratome (Leica; VT1000S). Free-floating 45 μm thick coronal sections were collected serially. Alternate sections were incubated for 48 hours at 4˚C with one of the following antibodies diluted in PBS containing triton X-100 (0.2%; PBST): anti-NeuN (1:1000, Tamecula), rabbit anti-PV (1:800; Swant PV28), rat anti-SOM (1:500; Millipore MAB354), rabbit anti-NPY (1:8000, Sigma N9528), rabbit anti-VIP (1:500, ImmunoStar 20077), or rabbit anti-NOS1 (1:500; Santa-Cruz sc-648) antibodies. After washing in PBST, sections were incubated with AlexaFluor 568 goat anti-rabbit or AlexaFluor 568 goat anti-rat antibodies (1:300; Invitrogen). Sections were rinsed in PBST, mounted in Vectashield (Vector) containing Dapi.

## Quantification of GABAergic immunostained neurons

Immunostained slices of GAD67-GFP knock-in (Δneo) transgenic mice were observed with a fluorescent microscope (Zeiss, Axio Imager M1) equipped with an AxioCam MRm CCD camera (Zeiss). For each mouse and each marker, a ventral slice, a median slice and a caudal slice were selected. Mosaics were constructed for each slice from 10X magnification images spanning the hilus and granular cell layer (GCL) of the dentate gyrus using AxioVision 4.7 (Zeiss). Counts were performed using a procedure written in IGOR PRO 6 (WaveMetrics) adapted from Perrenoud at al., 2012 (S1A Fig). Briefly, the outlines of the GCL outer and inner blades, or of the hilar outer and inner halves, were traced with connected line segments. 31 points were positioned at regular intervals along each outline and matching points were connected with transversal line segments. Transversal segments were in turn divided into 8 equal parts to construct a grid. Adjacent quadrilaterals along the longitudinal direction of the grid divided the GCL or the hilus halves in 8 depth bins (S1A5 Fig). Reference scaling was estimated using scale bars provided by AxioVison 4.7 and was verified with a graticule. For each animal and each coronal position, final values of densities were computed, by normalizing the sum of counted cells in each bin by the corresponding area.

## Slice preparation for electrophysiological recordings

Experiments were performed in accordance with the guidelines of the European Community Council Directive of November 24, 1986 (86/609/EEC). For recording performed in the dentate gyrus of the hippocampus, C57BL/6 mice (Charles River) aged 2 to 3 months were deeply anesthetized with ketamine (1%) and xylazine (1‰) and were perfused intracardially with an ice-cold sucrose solution containing (in mM): CaCl2 (1), glucose (10), KCl (1), MgCl2 (5), NaHCO3 (26), sucrose (248). Mice were decapitated and brains were removed in the same ice-cold solution. Horizontal slices, 300 μm thick, were cut with a vibratome (VT-1200, Leica) and put for 10 minutes in a chamber containing artificial Cerebro-Spinal Fluid (aCSF) warmed to 33˚C containing (in mM): CaCl2 (2), KCl (2,5), MgCl2 (3), NaH2PO4 (1), NaCl (124), glucose (11); NaHCO3 (26,2). Slices were then kept at room temperature until use. For recording performed in the primary somatosensory cortex, juvenile C57BL/6 mice (Janvier) aged P14–P17 were deeply anesthetized with halothane and decapitated. Brains were quickly removed and cut into 300 μm thick slices with a 30–40˚ inclination from the sagittal plane into an ice-cold slicing solution containing (in mM): choline chloride (110), sodium ascorbate (11.6), MgCl2 (7), KCl (2.5), NaH2PO4 (1.25), glucose (25), NaHCO3 (25), and sodium pyruvate (3.1). Slices were maintained at room temperature until use in a holding chamber containing artificial cerebrospinal fluid (aCSF) containing (in mM): CaCl2 (2), KCl (2.5), MgCl2 (1), NaH2PO4 (1.25), NaCl (126), glucose (20), NaHCO3 (26) and kynurenic acid (1). All solutions were continuously aerated with Carbogen (95%O2/5%CO2) (Air Liquide).

## Whole cell patch-clamp recordings

Slices were submerged in a recording chamber, placed on the stage of an Axioskop 2FS microscope (Carl Zeiss), equipped with Dodt gradient contrast optics (Luigs & Neuman), and a CoolSnap FX CCD camera (Roper scientific) and visualized by using infra-red (IR) videomicroscopy. The preparation was continuously superfused (1–2 ml/min) with oxygenated aCSF. Pipettes (2 to 6 MΩ) were pulled from borosillicate capillaries and filled with 8 μl of autoclaved internal solution containing 144 mM K-gluconate, 3 mM MgCl2, 0.5 mM EGTA, 10 mM HEPES, pH 7.2 (285/295 mOsm), and 3 mg/mL biocytin (Sigma). Whole cell recordings were performed at room temperature in the current Clamp mode of a MultiClamp 700B amplifier (Molecular Devices). Signals were filtered at 4 kHz and digitized at 50 kHz using an analog signal converter (Digidata 1322A, Molecular Devices) connected to a computer running pClamp 10.2 (Molecular Devices). Junction potentials were not corrected. Correction would shift the values of the resting membrane potential of hilar and cortical neurons by -14.8 mV and -15 mV respectively (online data). Resting membrane potential (RMP) was not used to compare hilar and cortical neurons, as detailed below.

## Single cell RT-PCR protocol

Recordings were kept under 10 min to limit the degradation of transcript due to cytoplasm dialysis. At the end of recordings, the cytoplasm was gently aspirated into the patch pipette which was then slowly removed to allow closure of the cell membrane. Pipette contents were expelled into test tubes in which reverse transcription was performed overnight at 37˚C as previously described [49]. Products of reverse transcription were stored at -80˚C until further processing. The single cell RT-PCR protocol was designed to detect mRNAs coding for the vesicular glutamate transporter 1 (VGluT1), the two isoforms of glutamic acid decarboxylase (GAD65 and GAD67), the neuronal nitric oxide synthase (NOS1), the calcium binding proteins calretinin (CR) and parvalbumin (PV), and the neuropeptides somatostatin (SOM), vasoactive intestinal polypeptide (VIP), neuropeptide Y (NPY) and cholecystokinin (CCK) (Table 1). Two successive rounds of amplification were performed using nested primer pairs [43]. All markers were amplified in bulk with a first set of primers (Table 1), undergoing 21 PCR cycles (94˚C for 30s, 60˚C for 30s and 72˚C, 35s) in a final volume of 100 μl. Markers were then amplified separately for 35 additional PCR cycles using a second primer pair situated inside the amplicon of the first primer pair (nested primer pairs). All primers (Table 1) were designed to be on two different exons of the target mRNA to exclude genomic contaminations. The presence of products of amplification was detected on a 2% agarose gel stained with ethidium bromide (Sigma).

## Electrophysiological analysis

Electrophysiological properties were quantified using 16 parameters measured on current clamp recordings of the membrane potential (Vm) in response to 800ms current pulses. Resting Membrane Potential (RMP) (**1**) was measured immediately after gaining intracellular electrical access. Input resistance (Rm) (**2**), Membrane time constant (τm) (**3**) and Membrane capacitance (Cm) (**4**) were determined in response to an hyperpolarizing current pulse eliciting a maximal voltage response of –10 to –15mV [50]. Rm was calculated using Ohms law. τm was computed as the time constant of a single exponential fit of the voltage response from onset to maximum hyperpolarization. Cm was obtained following the formula Cm = τm/Rm. Some neurons undergo a partial repolarization following hyperpolarization peak reflecting the activation of the voltage activated cationic current (Ih). We quantified this property using the current-voltage (IV) relationship in current injections ranging from –100 pA to 0 pA with 10pA increments. The maximal resistance $R_{hyp}$ and steady state resistance $R_{sag}$ were estimated

**Table 1. PCR primers.**

| Genes | | First PCR primers | Size | | Second PCR primers | Size |
|---|---|---|---|---|---|---|
| GAD67 | sense | ATGATACTTGGTGTGGCGTAGC | 253 | sense | CAATAGCCTGGAAGAGAAGAGTCG | 177 |
| NM_008077.2 | antisense | GTTTGCTCCTCCCCGTTCTTAG | | antisense | GTTTGCTCCTCCCCGTTCTTAG | |
| GAD65 | sense | CCAAAAGTTCACGGGCGG | 375 | sense | CACCTGCGACCAAAAACCCT | 248 |
| NM_008078.1 | antisense | TCCTCCAGATTTTGCGGTTG | | antisense | GATTTTGCGGTTGGTCTGCC | |
| CR | sense | TTGATGCTGACGGAAATGGGTA | 265 | sense | GCTGGAGAAGGCAAGGAAGG | 151 |
| NM_007586.1 | antisense | CAAGCCTCCATAAACTCAGCG | | antisense | ATTCTCTTCGGTCGGCAGGAT | |
| PV | sense | GCCTGAAGAAAAAGAACCCG | 275 | sense | CGGATGAGGTGAAGAAGGTGT | 163 |
| NM_013645.2 | antisense | AATCTTGCCGTCCCCATCCT | | antisense | TCCCCATCCTTGTCTCCAGC | |
| SOM | sense | ATGCTGTCCTGCCGTCTCCA | 250 | sense | GCATCGTCCTGGCTTTGGG | 170 |
| NM_009215.1 | antisense | GCCTCATCTCGTCCTGCTCA | | antisense | GGGCTCCAGGGCATCATTCT | |
| NPY | sense | CGAATGGGGCTGTGTGGA | 297 | sense | CCCTCGCTCTATCTCTGCTCGT | 220 |
| NM_023456.2 | antisense | AAGTTTCATTTCCCATCACCACAT | | antisense | GCGTTTTCTGTGCTTTCCTTCA | |
| NOS1 | sense | CCTGGGGCTCAAATGGTATG | 373 | sense | CCTGTCCCTTTAGTGGCTGGTA | 236 |
| NM_008712.1 | antisense | CACAATCCACACCCAGTCGG | | antisense | GATGAAGGACTCGGTGGCAGA | |
| VGluT1 | sense | CCCTTAGAACGGAGTCGGCT | 593 | sense | ACGACAGCCTTTTGCGGTTC | 367 |
| NM_182993.1 | antisense | TATCCGACCACCAGCAGCAG | | antisense | CAAAGTAGGCGGGCTGAGAG | |

using the slope of linear fits to the IV curves at maximal and steady state hyperpolarization [27]. Sag Ratio (**5**) was computed as $(R_{sag} * 100) / R_{hyp}$. Rheobase (**6**) was defined as the minimal current necessary to induce an action potential. At high firing frequencies, adaptation adopts a complex kinetic characterized by an early exponential and late linear decrease in firing frequency. To capture this, we used the spike train induced by the maximal depolarizing current before saturation and fitted the Inter-spike intervals with the function: $F_{sat} = A_{sat}*exp(-t/\tau_{sat}) + m_{sat}*t + F_{ma}$ where $A_{sat}$ is the Amplitude of late adaptation (**7**), $\tau_{sat}$ the Time constant of late adaptation (**8**), $m_{sat}$ is the Maximal steady state frequency (**9**) and $F_{max}$ is the Late adaptation (**10**) [51]. To quantify spike waveforms, we used 6 parameters measured on the train evoked by the minimal current injection eliciting more than 2 action potentials. Spike amplitude (**11**) and Spike duration (**12**), After Hyperpolization Potential (AHP) amplitude (**13**) and AHP latency (**14**) were measured on the first action potential. Spike amplitude was defined as the voltage between spike onset and peak. Spike duration was computed as the spike width at half amplitude. AHP amplitude and latencies were calculated relative to the onset of the action potential [52]. Spikes can also display variations in amplitude and duration at half width within a spike train. Thus, Amplitude reduction (Amp.Red) (**15**) and Duration increase (Dur.Inc) (**16**) were computed as (A1-A2)/A1 and (D2-D1)/D1 where A1, D1, A2 and D2 are the amplitude and duration of the first and second action potentials [44].

## Data analysis and statistics

Data analysis and statistics were performed in the Matlab environment (MathWorks, Natick, Massachusetts).

Unsupervised clustering: In addition to the 16 electro-physiological parameters described above, the parameters retained for clustering also included the expression of VGluT1, GAD, NOS1, PV, CR, NPY, SOM and CCK. Molecular parameters were represented as boolean variables. GAD corresponds to the expression of GAD65 or GAD67. Parameters were z-scored to ensure that they had equal weight on the clustering. A first estimate of the cluster was obtained using Ward's method. Then, a second clustering was performed using the K-means algorithm initialized on ward's cluster centroids (as previously described [47]). Clustering quality was

quantified using silhouette analysis. The silhouette value S(i) is computed for each neuron i as: S
(i) = (b(i)-a(i))/max[a(i),b(i)] where, a(i) is the average distance between i and the neurons of the
same cluster and b(i), the average distance between i and the neurons of the closest cluster. A posi-
tive silhouette value indicates that a neuron is closest to the neurons of its own cluster. By contrast,
a negative value indicates a potential misclassification. The impact of each parameter on clustering
quality was assessed by scrambling (i.e. permuting values across neurons [27]). Scrambling dis-
rupts the correlation between parameters without affecting their distributions. Scrambling was
performed 1000 times per parameter. Parameters were considered significant if scrambling
resulted in a decrease in mean silhouette value across neurons more than 95% of the times.

**Statistical comparisons.**   For electrophysiological variables, pairwise differences between
identified clusters were tested using nonparametric Mann-Whitney U-tests. Statistical differ-
ence in the expression of molecular markers were estimated using binomial tests. Pairwise
association between markers were tested directionally i.e. by testing for a difference in the
expression of marker A between cells expressing marker B and cells not expressing marker B
and vice versa (S1 Table).

Permutation test of correlational structure for marker expression: To test for an overall cor-
relational structure in marker expression within clusters, we devised the following permuta-
tion test. We reasoned that if markers expression is uncorrelated this should translate in a low
variance of the eigenvalues of their covariance matrix (the eigenvalues of a covariance matrix
can be thought of as the magnitude of the components in principal components analysis
(PCA)). Conversely, if there is a strong correlation structure in the marker expression, a lot of
the multiparametric population variance should be captured by a few components and the var-
iance of eigenvalues should be high. We thus scrambled the values of each molecular marker
to disrupt their correlation structure. This operation was repeated 10000 times to draw a distri-
bution of eigen value variance under the null hypothesis (H0). We concluded that the correla-
tion structure between markers was significant if the real eigen value variance was superior to
95% or more of the H0 distribution.

Resampling test of principal component alignment and centroid position: We wanted to test
whether two samples (i.e. neurons) defined in a multiparametric space come from the same
overall population. We reasoned that if it is so, the two samples should have similar values for
metrics depending on the overall shape of their distribution in the multiparametric space. We
thus selected two such metrics: 1) the angle between the first principal components (PC1) of
each of the two samples and 2) the Mahalanobis distance between their centroids. Mahalonobis
distances can be thought of as an analog of the z-score in multi-dimensional spaces. One unit of
distance indicates that two points are one unit of covariance away. Here distances were esti-
mated with respect to the covariance matrix of cortical interneurons. To estimate how these
metrics are distributed under the null hypothesis, we took advantage of the fact that, in our case,
one sample (i.e. cortical neurons, n = 268) was much larger than the other (i.e. hilar neurons,
n = 51). This made it possible to estimate a distribution of PC1 angle and centroid distance by
resampling as many observations as contained in the smallest sample (i.e. hilar interneurons)
from the largest sample (i.e. cortical neurons). Resampling was performed 10000 times.
Values were considered significant when superior to the 95th percentile of the resampled
distribution.

## Results

### GAD expression among neurons of the hilus and granular cell layer

In the telencephalon, GABAergic interneurons diversify into families defined by the expres-
sion of classical molecular markers such as the calcium binding protein parvalbumin (PV) and

the neuropeptides Somatostatin (SOM), Vasoactive Intestinal Peptide (VIP) and Neuropeptide Y (NPY). Relevant markers also include the calcium binding protein CR, the neuropeptide CCK and the enzyme neuronal NO synthase NOS1 [26,28,30,31,53]. In this study, we investigated the relevance of these markers for the classification of GABAergic interneurons in the hilus of the dentate gyrus. We therefore first set out to characterize their distribution across hilar neurons using immunohistochemistry.

Histological studies relying on GABA or Glutamate Acid Decarboxylase (GAD) immunolabeling are prone to under-detection. For this reason, reports on the distribution of GABAergic interneurons in the dentate gyrus have not led to a converging conclusion [54–59]. To overcome this difficulty, we took advantage of the glutamic acid decarboxylase (GAD)-67 GFP knock-in mice which selectively express GFP in GABAergic neurons [42]. We immunostained the neuronal nuclear antigen NeuN to visualize all neurons and quantified the distribution of GABAergic neurons in the hilus and dentate gyrus. We delineated four regions of interest: the granule cell layer (GCL) inner and outer blades and the hilar inner and outer halves (S1 Fig). Each region was then divided into 8 laminar bins (S1A5 Fig; Materials and methods) in which the overall neuronal density was evaluated by normalizing cell counts by bin area (S1B–S1E Fig). Counting of NeuN-expressing cells was restricted to the hilus as cell densities were too high in the GCL and individual cells could not be distinguished. To detect potential variations in cell density among hippocampal regions, counts were repeated at three distinct positions along the antero-posterior axis of the dentate gyrus: rostral, median and caudal. We found that NeuN-labeled cells were evenly distributed throughout the hilus (S1 Fig). However, GABAergic neurons were preferentially located near the subgranular layer (S1B–S1E Fig). GABAergic cells represented 40% of hilar neurons (S1F Fig).

## Distribution of the classical markers CR, PV, SOM, NOS1, NPY and VIP in the hilus and granular cell layer GABAergic neurons

We then analyzed the expression of classical markers of telencephalic interneurons in the hilus and dentate gyrus of GAD67-GFP mice. The GCL was excluded from counting for CR-immunolabeled cells as CR is transiently expressed in the adult-born granule cells of the subgranular cell layer [60].

In the hilus, CR is expressed in glutamatergic mossy cells, which represent most, if not all, non-GABAergic hilar neurons [14,61–64]. Accordingly, we found that the colocalization of CR and GABA was sparse (Fig 1A1). Non-GABAergic CR positive neurons were densely distributed throughout the hilus, confirming that CR is mainly expressed in excitatory cells in this region. CR neurons were denser in the rostral hilus compared to median and caudal hilus (S2A Fig). GABAergic neurons expressing PV were found at all depths of the GCL and hilus but were more concentrated near the GCL-hilus border (Figs 1B and S2B). PV neurons represented 43% of GABAergic neurons in the GCL and 22% in the hilus. Conversely, the density of SOM-expressing neurons was higher away from the GCL (Figs 1C and S2C). About 80% of SOM hilar neurons were found in the deep hilus (bins 10–21). SOM expressing neurons represented 27% of GABAergic neurons in the hilus and 8% in the granular cell layer. NOS1 and NPY expression followed the distribution of the overall GABAergic population (Fig 1D and 1E). 35% of GABAergic neurons were immunolabeled for NOS1 in the hilus and 31% in the GCL. The density of NOS1 neurons was lower in the caudal hilus (S3A Fig). NPY was expressed in 60% of hilar GABAergic neurons and 31% of GABAergic neurons expressed NPY in the GCL (S3B Fig). Finally, we found that VIP was virtually not expressed in the GCL and hilus. VIP labeled neurons accounted for less than 4% of GABAergic cells across the dentate gyrus (online data).

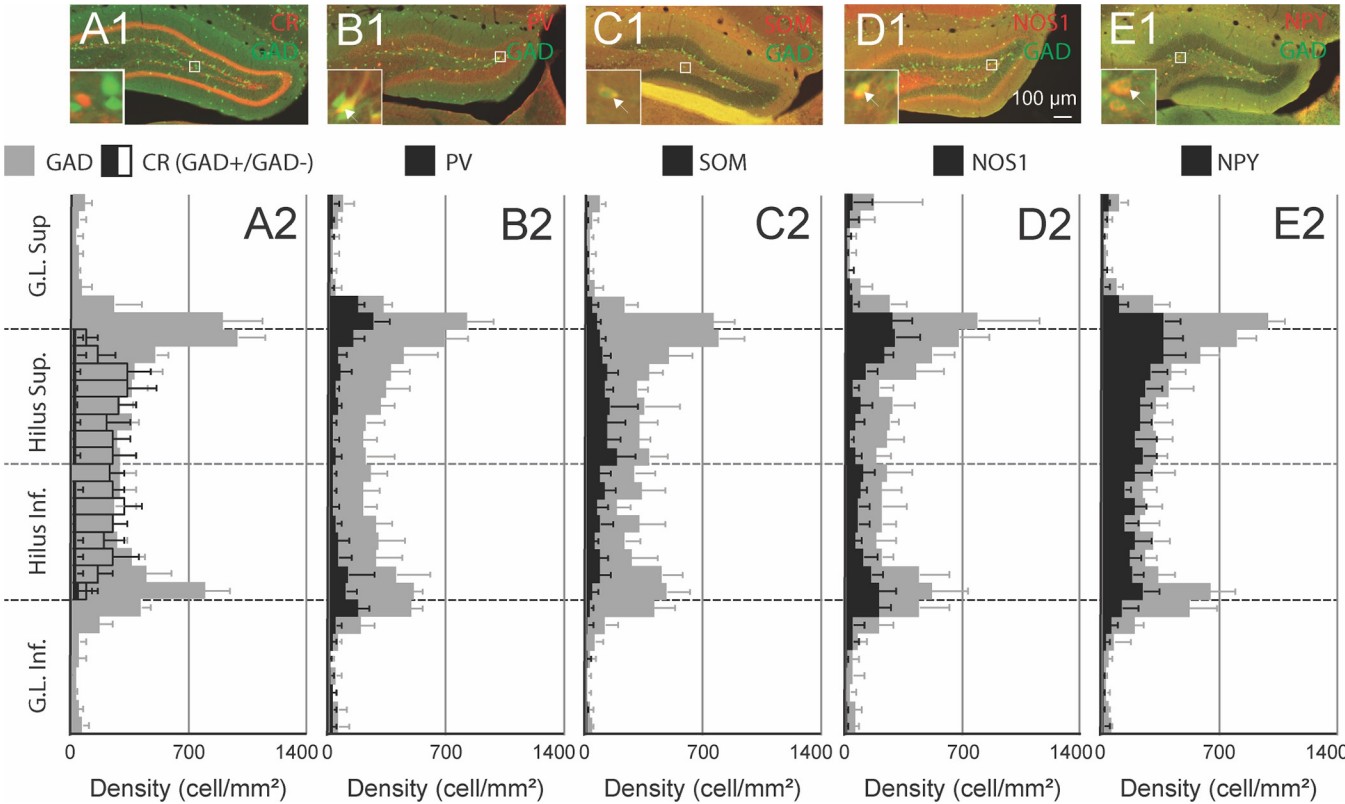

**Fig 1. Immunoreactivity for classic GABAergic interneuron markers is observed within specific subregions of the dentate gyrus.** A1-E1: CR, PV, SOM, NOS1 and NPY immunostaining in the dentate gyrus of a GAD67-GFP knock-in mouse, the delineated area is enlarged in the inset. A2-E2: Densities of GFP expressing cells (grey), and PV, SOM, NOS1 and NPY positive GFP expressing cells (black) in the outer blade of the granular cell layer (G.L. Sup: bin 1 to 8), upper half of the hilus (Hilus Sup.: bin 9 to 16), lower half of the hilus (Hilus Inf.: bin 17 to 24) and inner blade of the granular cell layer (G.L. Inf.: bin 25 to 32). In A1, the black outline represents the density of all GABAergic and non-GABAergic CR positive cell. CR positive cells were not counted in bin 0 to 7 and 24 to 31. PV and NOS1 expressing GABAergic interneurons accumulate near the border between the hilus and the granular cell layer, whereas SOM expressing cells are more concentrated in the hilus (n = 7 mice/marker, error bars: sem).

## Clustering of hilar GABAergic neurons based on electrophysiological profile and classical markers

Immunohistochemical data indicate that PV, NOS1 and SOM are expressed in GABAergic neurons located in specific territories of the hilus. This suggests that classical markers might be useful to delineate classes of GABAergic neurons, as is the case in other telencephalic regions. To test this hypothesis, we performed patch-clamp recordings combined with scRT-PCR on hilar neurons. Sixteen electrophysiological parameters were measured, in accordance with the Petilla nomenclature [27,32,47]; Material and methods). Our scRT-PCR protocol was designed to detect the expression of 10 molecular markers (Material and methods): Vesicular Glutamate Transporter 1 (VGluT1), Glutamic Acid Decarboxylase 65K and 67K (GAD65 and GAD67), CR, PV, SOM, NOS1, NPY, VIP and CCK. In order to prevent mRNA degradation from cytoplasm dilution, the time of whole cell was limited to less than 10 minutes. Primers were targeted to different exons to detect and exclude false positives resulting from genomic contamination. As previously reported [43,49], our protocol failed to detect genetic expressions on material harvested form the slice's extracellular milieu indicating that false positives due to extracellular contamination are extremely unlikely. To reduce false negatives caused by insufficient sample collection, cells expressing less than two markers were excluded.

However, the harvesting of part of the cytoplasm and the short recording duration inherent to the single cell RT-PCR technique prevent a good diffusion of neuroanatomical tracer such as biocytin into the axonal arbor. Consequently, the axonal projections of recorded neurons were not recovered.

We harvested 147 hilar neurons. This sample showed a good concordance with histological results and previously published data supporting our approach. Ninety-six neurons expressed VGlut1, together with CR (60.4%) and CCK (56.3%), which is characteristics of glutamatergic mossy cells. According to our immunohistochemical data, we also found that VIP was rarely expressed in hilar neurons (2 out of 147). This marker was thus not included in the analysis.

We found 51 GABAergic neurons in our sample (i.e. expressing GAD65 and/or GAD67). To understand how they diversify, Ward's unsupervised clustering was performed based on the electrophysiological and molecular properties [44,51,52,65–69]; Material and methods). This approach groups neurons into a dendrogram (i.e. a hierarchical tree) based on similarity (Fig 2A; Material and methods). The length of the tree branches represents the distance between groups of neurons. To delineate clusters in the dendrogram, we set a cutoff at the maximal distance step in the tree (Thorndike's method; Thorndike 1953; Figs 2 and S4). This identified 3 main clusters. Ward's method can sometimes miss assign observations during the iterative process, resulting in overlapping clusters. To correct for this, we used K-means algorithm initialized on Ward's clusters centroids [27,47]. This only resulted in the reassignment of one cell (Fig 2B). Clustering quality was then assessed by performing a silhouette value analysis (see Materials and Methods, Fig 4C). The mean silhouette value across GABAergic neurons was 0.294. Positive values of silhouette were observed for all neurons, validating our partition of the data (Fig 2C). To gain insight about the distribution of clusters in the multi-parametric space, we used principal component analysis (PCA). The projection of our sample along the first and second order principal components indicated a strong separation between cluster 1 and the rest of the data (Fig 2D). However, clusters 2 and 3 seemed to lie in closer continuity. Together the first and second principal components accounted for 37.2% of the sample variance.

To understand how electrophysiological and molecular features impact the clustering, we assessed the contribution of each parameter with scrambling [27], Material and methods, Fig 2E). Scrambling resulted in a significant decrease in clustering quality for 10 out of 16 electrophysiological parameters. Surprisingly however, only 3 markers of interneuron significantly impacted clustering: CR, NPY and SOM (together with the glutamatergic marker VGlut1). To determine the contribution of each type of feature to clustering, we repeated our clustering approach using only specific subsets of parameters (i.e. (1) electrophysiological, (2) molecular, (3) all significant, (4) significant electrophysiological and (5) significant molecular parameters, Fig 2F). Using all significant parameters (Fig 2E) resulted in the same grouping of interneurons as with the full parameter set (Fig 2F, 4th column). This confirmed the relevance of these parameters and validated our clustering. Using electrophysiological or molecular parameters in isolation resulted in a decrease in clustering quality (mean silhouette, Electrophysiological: 0.234, Molecular: 0.104) and in the reassignment of 14 and 12 neurons respectively (Fig 2F, 2nd and 3rd column). This indicates that neither type of feature fully recapitulates the diversity of hilar GABAergic interneurons, as is the case in other telencephalic regions [32]. Interestingly, using only significant electrophysiological parameters resulted in the widest disruption of clustering (Fig 2F, 5th column, 15 reassigned cells). However, clustering based exclusively on the four significant molecular markers VGlut1, CR, SOM and NPY resulted in the reassignment of only 5 neurons. Thus, it appears that the expression of these four markers recapitulates much of the diversity of hilar GABAergic neurons.

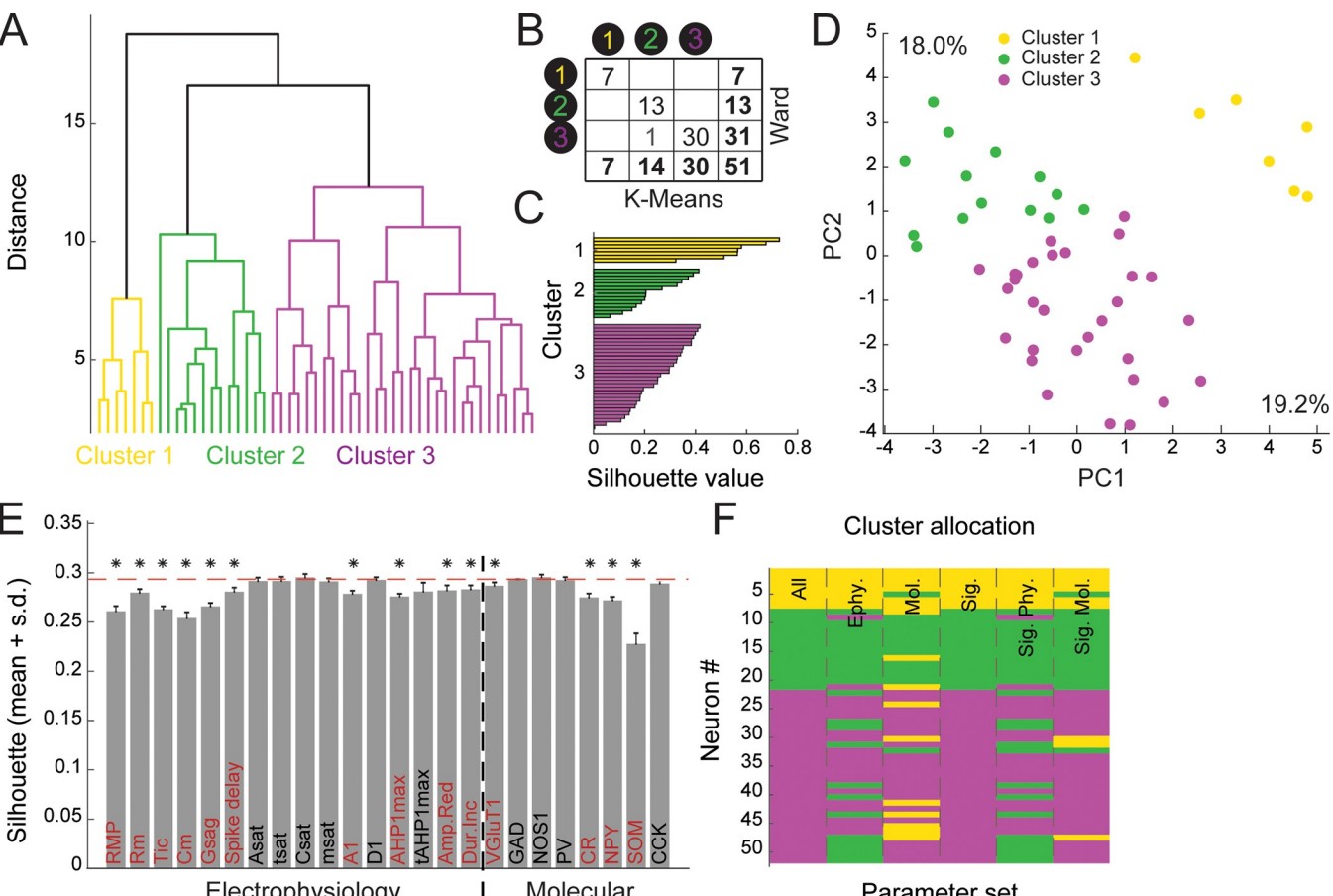

**Fig 2. Hilar GABAergic neurons segregate into 3 main clusters. A:** Ward's unsupervised clustering applied to 51 hilar GABAergic neurons based on electrophysiological and molecular properties characterized with single cell RT-PCR. 16 electrophysiological and 8 molecular parameters were used. The analysis disclosed 3 clusters of cells. The x-axis represents the individual cells and the y-axis the distance of aggregation. **B:** Matching table of Ward clusters and clusters generated by the K-means algorithm initialized on Ward clusters centroids. The K-means algorithm corrects cells miss-assigned by Ward method generating non-overlapping clusters. **C:** Cluster silhouettes plot: Each individual cell is represented by a horizontal line. No negative silhouette values were observed, indicating no potential miss-assignment. **D:** Two-dimensional projection of hilar GABAergic neurons on the first and second principal components (PC) following Principal Component Analysis. Percentages beside the x and y axes indicate the fraction of the sample variance explained by the first and second principal components respectively **E:** Impact of electrophysiological and molecular parameters on clustering quality (see Methods for parameter details). The values of each parameter were shuffled 1000 times. Parameters were considered significant (*; red label) when shuffling resulted in a decrease in mean silhouette value more than 95% of the times. **F:** Cluster allocation is robust to parameter sets. Allocation of each neuron (y-axis, color code as in A) using the following parameters sets (x-axis from left to right): 1. All, 2. Electrophysiological, 3. Molecular, 4. Significant, 5. Significant Electrophysiological, 6. Significant Molecular.

## Properties of hilar GABAergic interneuron clusters

We then examined the properties of each cluster of hilar GABAergic interneurons. Cluster 1 consisted of 7 cells characterized by their high single-cell detection rate of mRNAs for VGluT1 (85.7%), CR (85.7%) and CCK (71.4; Fig 3A). At the electrophysiological level, these cells displayed strong inward rectification in response to hyperpolarizing current pulses and long firing latency at rheobase (Fig 3B). They showed higher resting membrane potential (RMP: -51.1 +- 1.5 mV, >Cluster 2, p = 0.035) and longer spike duration (D1: 1.30 +- 0.08 ms, >Cluster 2, p = 0.040) than other neurons (S5 Fig). These features were reminiscent of the properties of hilar glutamatergic mossy cells. Thus, the neurons in cluster 1 could correspond to a subset of mossy cells expressing GAD65 or GAD67 along VGlut1. To verify this, we performed a cluster analysis of hilar GABAergic neurons with our full sample of hilar glutamatergic neurons (S6

Fig). We found that most of the cells in cluster 1 (6 out of 7) segregated with glutamatergic neurons, confirming that they likely correspond to mossy cells expressing small amounts of GAD enzymes.

Cluster 2 neurons included 14 neurons expressing NPY (100%) and SOM (100%, Fig 3A). These neurons tended to display high input resistance (Rm: 480.6 +- 52.3 MOhms), membrane time constant (tic: 90.730 +- 8.124 ms) and membrane capacitance (Cm: 200.0 +- 14.3 pF). They also often exhibited delayed firing at rheobase ($1^{st}$ spike delay: 534.0 +- 37.4 ms) and a high amplitude of adaptation at saturation (Asat: 35.6 +- 9.3 Hz, >Cluster 3 (p = 0.040)).

Finally, cluster 3 neurons consisted of 30 neurons expressing various classical markers of interneuron, at low single-cell detection rates (Fig 3A) and displaying a variety of firing patterns (Fig 3D). The most widely expressed markers in cluster 3 were CCK (53.3%), NPY (33.3%), PV (30.0%) and NOS1 (30.0%). Marker expression did not seem associated with specific electrophysiological subtypes (see Fig 3D1–3D2 for an example of 2 neurons having similar electrophysiological properties and expressing PV and NOS1 respectively). To understand the diversity of cluster 3 neurons, we tested the association between the expression of all pairs of classical interneuron markers (Material and methods). No marker pair was co-expressed above chance level (S1 Table).

This suggested that the expression of classical interneuron markers in cluster 3 neurons was statistically independent and did not display any correlational structure. To test this hypothesis, we designed a test based on principal component analysis to search for the presence of correlational structure in a multiparametric sample (Material and methods). This test relies on the idea that if a strong correlational structure exists between a set of variables, a large fraction of the sample's variance should be summarized by a few principal components having high eigen values. Conversely, in the absence of correlational structure, principal components will tend to align with parameters and their eigenvalues should exhibit similar magnitudes (Fig 3E1). Following this idea, we reasoned that the variance of the principal components (PC)'s eigenvalues can serve as a statistic to test for the presence of correlational structure. The distribution of this statistic under the null hypothesis was estimated by scrambling parameters 1000 times (Fig 3E2, scrambling disrupts correlational structure while keeping the distribution of parameters intact). When performing this procedure on classical interneurons markers in cluster 3, we found that the variance of PC eigenvalue was not different from H0 (Fig 3E2). Thus, our analyses indicate that classical interneurons markers are expressed independently in cluster 3 neurons. However, a strong correlational exists between markers in cluster 2 as demonstrated by the corresponding significant higher distribution of the PC eigen value variance.

## Hilar and cortical GABAergic interneurons do not belong to the same neuronal classes

In most telencephalic regions, PV expression defines a major class of interneurons: the fast-spiking basket cell [26,28,30,31,53]. However, surprisingly, our clustering does not suggest that PV expression has a significant impact on the grouping/classification of hilar GABAergic neurons. This would imply that molecular expression in hilar GABAergic neurons does not relate to electrophysiological properties in the same way than in other forebrain regions.

To test this possibility, we compared our sample of hilar GABAergic neurons to a sample of 300 cortical interneurons from the barrel region of the mouse primary somatosensory cortex. This sample was acquired using patch-clamp recordings on slices, combined with scRT-PCR with a protocol similar to the method used in the present study (Material and methods). Part of this sample was analyzed in previous work [47,70]. VIP is expressed in cortical interneurons where it defines a major subclass [26,28,43,44,71,72]. As VIP was only sparsely found in hilar

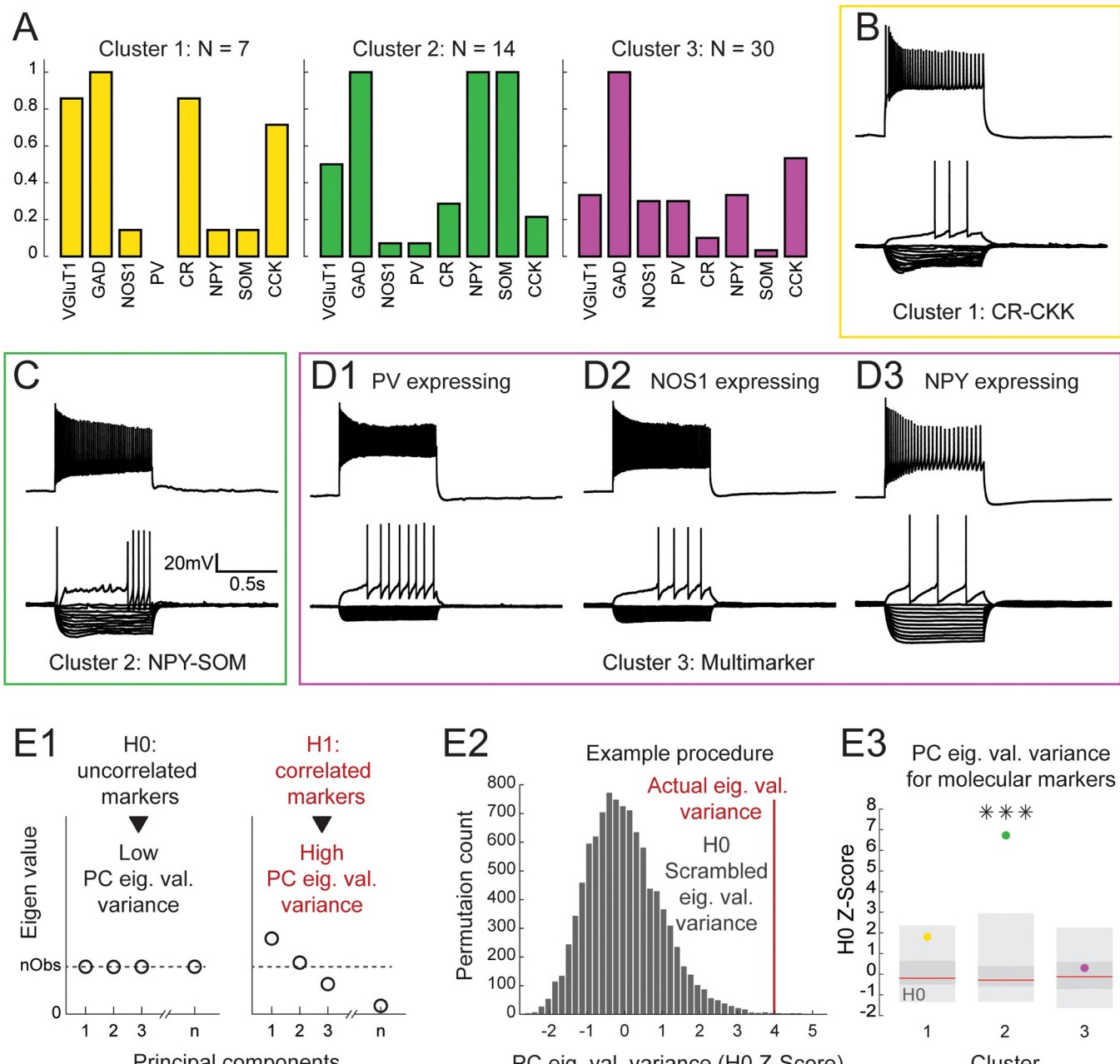

**Fig 3. Properties of hilar GABAergic interneurons. A:** Histograms showing the expression of molecular markers in cluster 1, 2 and 3. Cluster 1 neurons express VGlut1, CR and CCK and are probably a subset of hilar mossy neurons expressing GAD (see Figs 3 and S1). Cluster 2 neurons express NPY and SOM. Cluster 3 neurons express various marker including PV, NOS1 and NPY and CCK. **B:** Example of cluster 1 neuron current-clamp recording showing responses to current injections from -100pA to 0pA in 10pA increments and at rheobase (bottom), and near saturation (top). Scale as in C. **C:** Example of a cluster 2 neuron recording (presentation: same as B). **D1-D3:** Example of recordings of PV, NOS1 and NPY expressing cluster 3 neurons respectively (presentation: same as B). **E:** There is no correlational structure in the expression of markers in cluster 3. **1:** Sketch illustrating the H0 and H1 hypotheses in a Principal Component Analysis (PCA) based test of correlational structure between molecular markers. Under H0, markers are uncorrelated and the variance of the principal components (PC) eigen values is low. Under H1, several markers are correlated and the PC eigen values variance is high. **2:** The distribution of the PC eigen value variance under H0 was estimated by shuffling marker expression values 10000 times. **3:** Actual value of eigen value variance for markers association in cluster 1 and 3 (yellow, green and purple) overlaid on H0 distributions (light grey: 2.5–25% and 75–97.5% percentiles; dark grey: 25–75% percentiles; red line: median). Markers association is not significantly different from H0 in cluster 1 and 3.

interneurons, VIP-expressing neurons were excluded from this cortical sample, resulting in a set of 268 cells.

We next sought to determine whether hilar and cortical GABAergic interneurons subdivide in similar cell types. We thus repeated the unsupervised clustering approach described previously on our cortical sample. This identified four clusters of cortical neurons (Fig 4A). Cortical cluster 1 and 2 stemmed from the same branch in the dendrogram and consisted of adapting

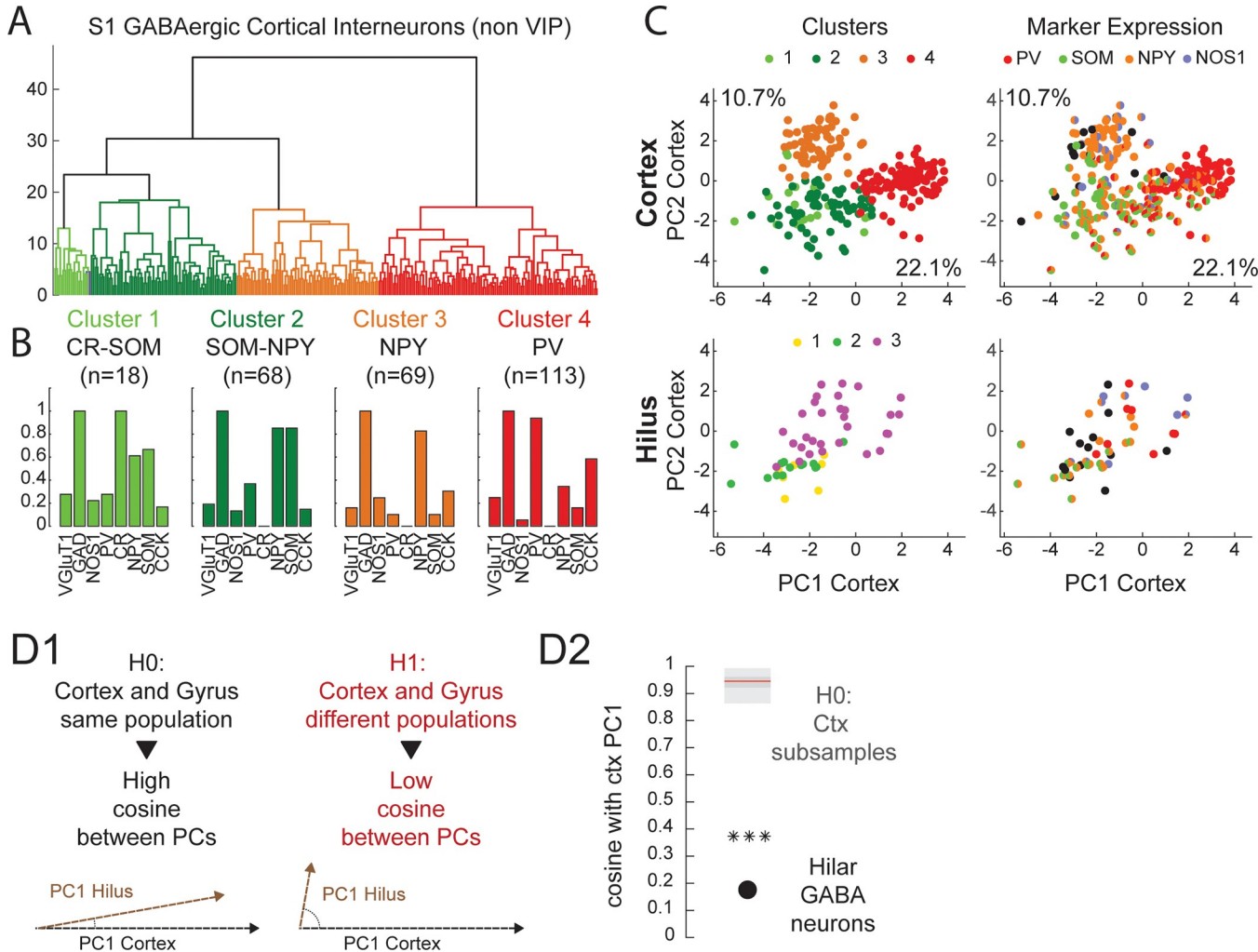

**Fig 4. Hilar GABAergic neurons do not diversify along the same axes as cortical interneurons. A:** Hilar GABAergic neurons were compared to a sample of 268 GABAergic interneurons recorded in the primary somatosensory cortex (S1). VIP neurons were excluded as this marker is not expressed by hilar GABAergic neurons. Ward's analysis based on the parameter used in Fig 2A segregated cortical interneurons in four clusters (x-axis: neurons; y-axis: distance of aggregation). **B:** Histograms of cortical interneurons expression of molecular markers. Cluster 1 neurons expressed CR and SOM, cluster 2 expressed SOM and NPY, cluster 3 expressed NPY and cluster 4 expressed PV. **C:** Clusters (left) and marker expression (right) of cortical (top) and hilar GABAergic neurons (bottom) projected on the two first principal components (PC) of the cortical population Percentages besides the x and y axes indicate the fraction of the cortical sample variance explained by the first and second principal components respectively. There is no obvious correspondence between clusters of GABAergic neurons in the gyrus and the cortex. **D:** The axes of maximum variance of cortical and hilar GABAergic neurons are not aligned. **1:** Sketch illustrating the H0 and H1 hypotheses in a Principal Component Analysis (PCA) based test of multiparametric variance alignment. Under H0, hilar and cortical GABAergic neurons come from the same population and their principal components (PC) have a high collinearity (cosine around 1). Under H1, hilar and cortical GABAergic neurons come from different populations and their principal components (PC) have a low collinearity (cosine around 0). The distribution of the PC cosines under H0 was estimated by subsampling 51 cortical neurons 1000 times. **2:** Actual value of PC cosine between cortical and hilar GABAergic neurons overlaid on H0 distributions (light grey: 2.5–25% and 75–97.5% percentiles; dark grey: 25–75% percentiles; red line: median). The alignment is significantly lower than expected from neurons sampled from the same population (***: p < 0.001).

neurons (msat: Cluster 1: -21.7 +- 3.3 Hz/s; Cluster 2: -26.74 +- 2.06 Hz/s) expressing SOM (66.7% and 85.3% respectively). In cluster 1 (18 neurons) SOM expression was associated with CR (100%), while it was associated with NPY in cluster 2 (85.3%; 68 neurons). Cluster 3 comprised 69 neurons expressing NPY (82.6%) but rarely other markers (Fig 3A). These neurons showed a marked adaptation (msat: -24.55 +- 1.48 Hz/s) and delayed firing at rheobase (1st spike delay: 283.8 +- 25.6 s), suggesting a neurogliaform type [26–28,73]. Finally, cluster 4 included 113 neurons expressing PV (93.8%) and having short action potentials (D1: 0.52 +- 0.01 ms) and high firing rates at saturation (CSat: 190.8 +- 4.1 Hz; S7 Fig) characteristic of a fast-spiking firing [74].

Cortical cluster 2 and hilar cluster 2 both contained adapting neurons expressing SOM and NPY suggesting a potential homology. However, no clear homologs could be found for cortical cluster 3 and 4 in our hilar population. To gain a better insight of the differences between hilar and cortical interneurons, we sought to compare their distribution on our multidimensional parameter space. We first investigated the differences in electrophysiological and molecular parameters between the sets of hilar and cortical GABAergic neurons expressing specific molecular markers. Selected markers included GAD (i.e. full set), PV, SOM and NPY (S8 Fig). We found that for all sets, there was a very significant difference in resting membrane potential (RMP) and firing rate at saturation (Csat). Hilar and cortical neurons were recorded in slightly different conditions (Material and methods). Thus, this was indicative of a potential population-wise bias. As such bias could artificially inflate the differences between cortical and hilar neurons, RMP and Csat were excluded from the parameter sets for subsequent analyses. This, however, had only a minimal impact on the main trends in our cortical and hilar GABAergic neurons samples (S9 Fig) and did not change the main conclusions of the study.

Parameter wise comparisons revealed 12 significant differences between hilar and cortical GABAergic neurons (Excluding RMP and Csat, S8 Fig). To visualize these differences, we performed principal components analysis on cortical interneurons and examined their distribution along their 2 main principal components (Fig 4C). Then, we projected hilar interneurons on the principal components of the cortical sample to get a visual impression of how hilar and cortical neurons overlap in space. Cortical clusters 1 (SOM-CR) and 2 (SOM-NPY) and hilar cluster 2 (SOM-NPY) occupied a similar region of the parameter space, confirming their potential homology. By contrast, hilar cluster 3 seemed to span across the territories occupied by cortical cluster 2 (SOM-NPY), 3 (NPY) and 4 (PV). Interestingly however, marker expression in hilar cluster 3 did not seem associated with specific subregions (Fig 4C, bottom-right). This confirmed that markers do not correlate well with other parameters in this cluster. It further indicated that hilar cluster 3 neurons do not have clear homolog in classical cortical classes.

## Hilar and cortical GABAergic interneurons diversify along distinct axes

Pairwise parameter comparison as well as visual inspection of PCA plots indicated that hilar and cortical interneurons do not diversify along the same axes. To give a stronger statistical foundation to that conclusion, we devised the following statistical test. We reasoned that if cortical and hilar interneurons were coming from similar populations and were keeping their phenotype, their first principal components (PCs) should be aligned. We thus computed the cosine of the two PCs to quantify this alignment. Cosines will have the value of 1 if the two PCs are aligned (i.e. the angle between them is close to zero) and zero if they are misaligned (i.e. orthogonal, Fig 4D1). To estimate the value distribution under the null hypothesis, we took advantage of the fact that our hilar sample (n = 51) was considerably smaller than our cortical sample (n = 268). This allowed to simulate an H0 distribution by resampling 51 neurons from

our cortical sample a set number of times (1000). We found that the cosine angle between the first PCs of hilar and cortical neurons was close to zero, indicating near orthogonality. This value was very significantly different from the expected cosine value under H0 (Fig 4D2). This strongly suggests that cortical and hilar interneurons do not diversify along the same axes.

## CCK, NOS1 and PV label different interneurons in the cortex and hilus

Our analysis of first PC cosine angle indicates that hilar and cortical GABAergic interneurons share only limited homology. This suggests caution in assuming that interneurons expressing particular markers in the hilus and in the rest of the forebrain are comparable classes. Here, the first PCs accounted for 23.8% and 22.1% of the variance hilar and cortical interneurons respectively. While this amount is substantial, it still leaves aside a large part of the information contained in our sample.

To gain additional insight into these specific differences, we compared the Mahalanobis distance between the centroids of hilar and cortical neurons expressing the markers GAD, NOS1, PV, NPY, SOM and CCK (Fig 5). Mahalanobis distances can be thought of as a multidimensional analog of the z-score. A distance of one indicates that two points are one unit of covariance away. Here, distances were calculated with respect to the deviation of cortical interneurons using all electrophysiological and molecular parameters (Fig 5A1). To estimate the distribution of distances under the null hypothesis, we once again resampled 51 cortical neurons a set number of times (1000). For all markers, we found that Mahalanobis distances were greater than expected under the null hypothesis (Fig 5A2). Distances were smaller for NPY and SOM (2.57 and 3.05 respectively). By contrast, distances were the greatest for CCK (4.77), NOS1 (4.00) and PV (3.90, Fig 5A2). To visualize the separation of cortical and hilar interneurons, we projected them on the first 2 PCs of the cortical population (Fig 5B). In addition, we examined their distribution on the axis connecting their centroids (Fig 5C). Once again, this analysis was performed for neurons defined by the expression of CCK, NOS1, PV, NPY, SOM. As expected, we found a substantial qualitative difference between the distribution of hilar and cortical neurons, for all markers.

Overall, our analyses suggest that hilar GABAergic interneurons diversify differently from forebrain interneurons. Moreover, they also indicate that classes defined on the basis of the expression of markers such as PV, NOS1 and CCK exhibit unique characteristics in the hilus of the dentate gyrus.

## Discussion

Hilar interneurons are usually classified based on axonal projection patterns and the expression of selected molecular markers. While it is often implicitly assumed that they separate into similar families to those of the rest of the forebrain, this assumption has never been directly tested. The analysis of certain electrophysiological and molecular characteristics permits a clear distinction between the major types of interneurons in the cortex [27,44,45,47,48,75] and hippocampus [46]. Therefore, in this study, we investigated whether electrophysiological and molecular criteria commonly used to classify GABAergic interneurons in the forebrain accurately differentiate hilar interneurons in the dentate gyrus.

First, we show that PV- and NOS1-expressing interneurons concentrate in the sub-granular zone whereas SOM-expressing interneurons are scattered in the center of the hilus. We then compare GABAergic interneurons sampled from the hilus and the somatosensory cortex, using patch-clamp recordings combined with single cell RT-PCR. While this feature set allows the separation of neocortical interneurons into well-established classes, it fails to yield a convincing partitioning of GABAergic hilar interneurons. Unsupervised clustering identifies 3

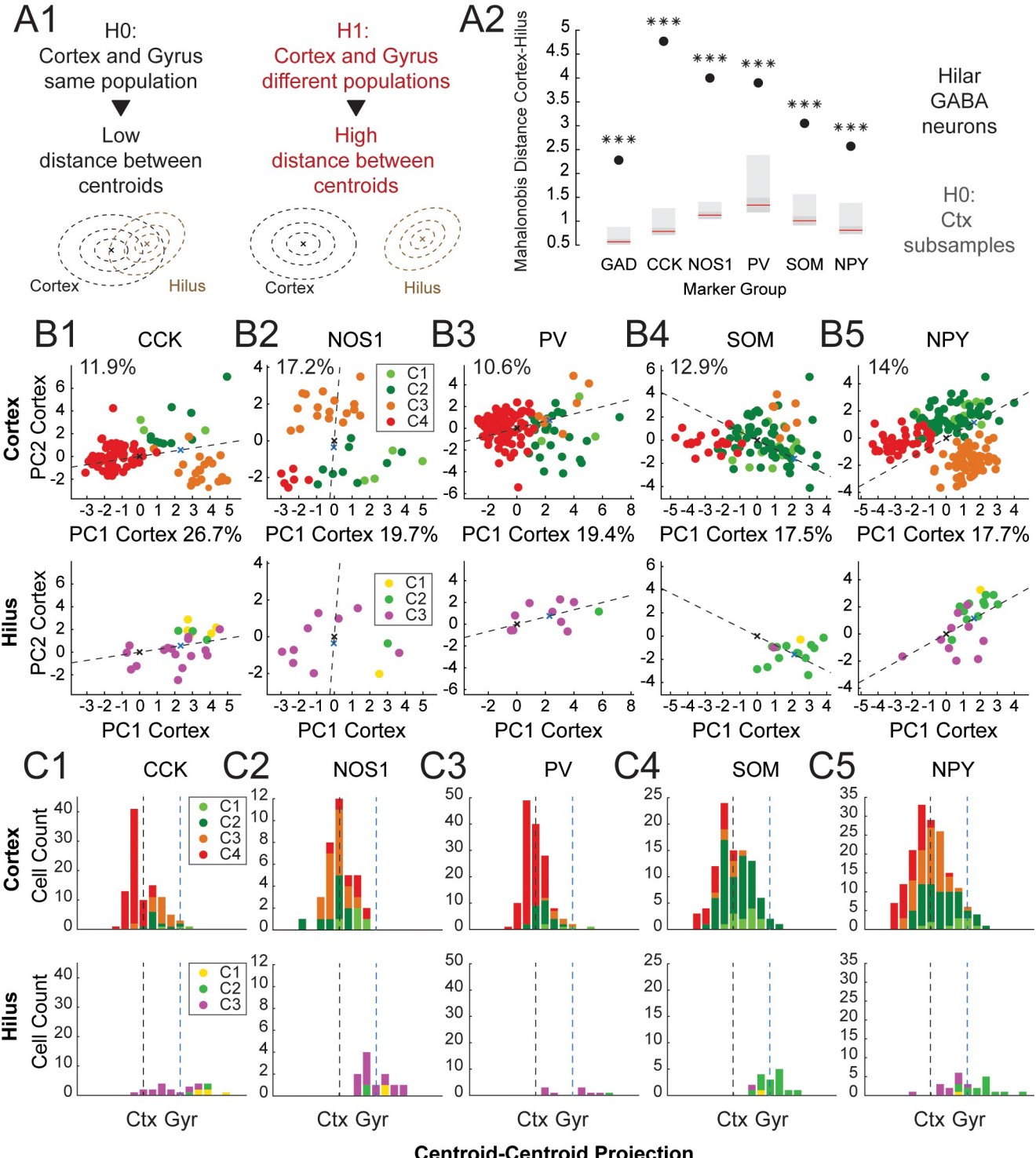

**Fig 5. Forebrain interneuron markers label distinct population in the cortex and dentate gyrus. A:** GAD, CCK, NOS1, PV, SOM and NPY expression defines populations having distinct centroids in the cortex and in the dentate gyrus. **1:** Sketch illustrating the H0 and H1 hypotheses in a test of distance between the centroids of 2 groups of neurons. Under H0, hilar and cortical GABAergic neurons come from the same population and their centroids are close. Under H1, hilar and cortical GABAergic neurons come from different populations and their centroids are further apart. The distribution of the centroid distance under H0 was estimated by subsampling as many cortical neurons as there were hilar neurons for each marker. Resampling was performed 1000 times. **2:** Actual value of Centroid-Centroid Mahalanobis distances between cortical and hilar GABAergic neurons for each marker overlaid on H0 distributions (light grey: 2.5–25% and 75–97.5% percentiles; dark grey: 25–75% percentiles; red line: median). Mahalanobis distances were

calculated with reference to the cortical population. Distances were highly significantly different from H0 for all markers (***: p < 0.001). **B:** Neurons expressing **1:** CCK, **2:** NOS1, **3:** PV, **4:** SOM and **5:** NPY in the cortex (**top**) and the dentate gyrus (**bottom**) projected on the first 2 principal components of the cortical population. Percentages in the x axis label and above the y axis indicate the fraction of the cortical sample variance explained by the first and second principal components respectively. The dotted line represents the axis linking the centroids of the cortical (black cross) and hilar (blue cross) populations. **C:** Histograms of the distribution of neurons expressing **1:** CCK, **2:** NOS1, **3:** PV, **4:** SOM and **5:** NPY projected on the axis linking the centroids of the cortical (black dotted line) and hilar (blue dotted line) populations in the cortex (**top**) and dentate gyrus (**bottom**).

groups: (1) putative mossy cells co-expressing GAD and VGlut1, (2) adaptive neurons expressing NPY and SOM and (3) a prominent cluster composed of heterogeneous interneurons expressing various markers at low frequency. Using a simple approach based on resampling, we provide statistical arguments to support the conclusion that hilar GABAergic cells display distinct profiles from those of cortical interneurons. Our results indicate that the electrophysiological properties measured are not strongly correlated with the expression of markers such as PV, CCK and NOS1 in hilar interneurons and suggest that hilar and cortical GABAergic cells share only limited homology.

## Hilar interneurons show limited homology with cortical GABAergic interneurons

Most GABAergic interneurons populating the neocortex, hippocampus and striatum share a common developmental origin [22–25]. Accordingly, they tend to include the same 3 major families: (1) soma and axo-axonic neurons showing fast discharge and expressing PV, (2) dendrite targeting interneurons expressing SOM and (3) neurogliaform neurons expressing NPY/ Reelin [26–28,30,31,53]. Regions of the cortex and hippocampus also include soma targeting interneurons expressing CCK and interneurons expressing VIP projecting to other types of inhibitory cells [22,28].

In the hilus of the dentate gyrus, different classification schemes of GABAergic interneurons have been proposed, based on morphological features, neuropeptides and calcium-binding proteins contents or electrophysiological behaviors [14,18–20,76–80].

Recent studies typically use transgenic lines to target interneurons expressing specific markers. Such studies have convincingly shown that PV- and SOM-expressing interneurons target the same principal neuron sub-compartments in the hilus than interneurons expressing these markers in the rest of the telencephalon [16,34–37,39,40]. However, few studies have attempted to establish a multiparametric classification of GABAergic hilar neurons. It has often been difficult in these attempts to show meaningful correlations between individual parameters, even leading some authors to speculate that hilar interneurons present a "continuum" of characteristics where each neuron is unique [20,21]. This raises the possibility that hilar GABAergic interneurons might have distinct properties than in the rest of the forebrain.

In the present study, we compare the relevance of commonly used electrophysiological properties and molecular markers for the grouping of GABAergic interneurons sampled from the hilus and primary sensory cortex, using patch-clamp combined with single cell RT-PCR. Our study thus takes a different approach to the classification of interneurons than usually used in the dentate gyrus and, to our knowledge, constitutes the first characterization of the co-expression for classical molecular markers in hilar neurons. Several steps were taken to ensure the accuracy of molecular detection. PCR primers were targeted to different exons allowing to detect genomic contamination, and aspiration of the extracellular milieu did not result in the spurious detection of genetic material. This makes false positive unlikely. Conversely, recording times were kept under 10 minutes and cells expressing less than 2 markers were excluded to limit false negatives. Our protocol has been successfully applied by our group

in the cortex of rat and mouse [43–45,47,75,81]. Thus, we are confident that the expression profiles reported here faithfully represent the pattern of expression of hilar interneurons.

Cortical and hilar samples were grouped based on a similar set of features using unsupervised clustering: a method that reduces personal bias and is widely used to implement neuronal classifications [27,44–47,65,67,75,82,83]. This approach successfully groups cortical interneurons into 3 well established classes: (1) PV expressing fast spiking neurons, (2) adapting neurons expressing SOM and (3) putative neurogliaform cells expressing NPY. However, it yields a markedly different partition of hilar GABAergic cells. We find that hilar neurons segregate into 3 clusters. One cluster corresponds to a subset of excitatory cells expressing a certain amount of GAD65 or GAD67 and showing features reminiscent of the properties of hilar glutamatergic mossy cells [63,84,85]. The other two clusters express one of the two GAD isoforms and should therefore correspond to the GABAergic interneurons that we sought to identify. The first of these clusters corresponds to adapting interneurons expressing NPY and SOM (15 neurons). They are thus potential homologs of SOM-expressing interneurons in the rest of the forebrain. The other group, which represents the majority of our sample (30 neurons out of 51), shows a variety of electrophysiological profiles and uncorrelated expression of molecular markers.

These disparities suggest that cortical and hilar GABAergic interneurons constitute distinct populations. To statistically test this idea, we compared two metrics summarizing the dispersion of our hilar and cortical samples in the multiparametric space: (1) the alignment between the first principal components and (2) the Mahalonobis distance between centroids (Mahalonobis distances are a generalization of z-scores in multi-dimensional spaces). In order to estimate the distribution of these metrics under H0 (i.e. the assumption that the two populations are identical), we took advantage of the fact that our samples of GABAergic interneurons are much larger in the cortex than in the hilus (268 against 51 cells). This allowed us to estimate the distribution of each metric under the assumption that hilar cells are a subsample of our cortical neuron set (Material and methods). The robustness of the present analysis is enhanced by the fact that the cortical cells were resampled 10,000 times. For both, we find that the observed values are highly unlikely under H0.

The statistical approach used in our study thus substantiates the conclusion that hilar and cortical interneurons only share limited homology. This approach is relatively straightforward and can be broadly applied to compare multidimensional samples when one sample is large enough to serve as a reference. To our knowledge, this is the first time that such an approach has been used to compare populations of neurons across brain regions. Our resampling approach can potentially be applied with any metric appropriately capturing aspects of a multi-dimensional distribution. We thus hope that our results will stimulate further research to compare neuronal populations across regions.

### Electrophysiological properties of hilar interneurons do not correlate well with the expression of commonly used molecular markers

In most forebrain regions, the expression of molecular markers correlates with specific electrophysiological profiles. For example, PV expression is associated with a fast-spiking neuronal type, while SOM neurons have adapting discharges and specific passive electrophysiological properties [26–28,30,31,53]. In the GABAergic hilar interneurons, markers such as PV and SOM are strongly associated to projection targets in the granular and molecular layers [16,34–37,39,40]. Markers like CCK and NOS1 have also been associated to a specific projection pattern [39,40]. However, how the neuronal expression of molecular markers correlates with specific cellular electrophysiological features remains unclear.

In the present study, we monitored the co-expression of 6 commonly used molecular markers (NOS1, PV, CR, NPY, SOM and CCK) together with electrophysiological properties in hilar neurons. Unexpectedly, we observed that these markers were not related to strongly distinctive electrophysiological properties [14,20]. How do our data impact the current understanding of hilar neurons diversity?

In our sample, 96 cells out of 147 could reliably be identified as excitatory, based on VGlut1 expression. As these neurons highly likely correspond to mossy cells [19,79,86], our data confirm that mossy cells constitute the main neuronal type of the hilus. These cells display low firing rate and are characterized by their high input resistance, high time constant and capacitance and strong inward rectification in response to hyperpolarized current steps [14,61–63,87]. Consistent with a previous report, these cells also show high expression of CR mRNA [88], together with low expression of other markers [14,61–63]. Our findings suggest for the first time, that mossy cells can express a certain level of mRNA for GAD enzymes. When cluster analysis was performed only on GAD-expressing neurons, we identified a cluster that likely corresponds to a subset of mossy cells expressing GAD65 or GAD67. It is important to stress however, that based on our data, the distinction between GABAergic and glutamatergic cells may be misleading. Indeed, VGluT1 was also detected in few expressing GAD neurons. However, our detection of VGluT1 mRNA in some GABAergic interneurons is consistent with its reported expression in GABAergic cortical interneurons [27,45,89] and also in several GABAergic cell types throughout the brain [90]. Nonetheless, the absence of VGluT1 immunoreactivity in symmetrical synapses [91] and the clear GABAergic phenotype of interneuron output in the dentate gyrus suggest a low level of VGluT1 expression in these interneurons.

In line with previous studies, we confirm that NPY and SOM are extensively co-expressed in hilar GABAergic interneurons and are evenly distributed throughout the hilus [14,92–98]. Furthermore, we find that SOM-NPY neurons constitute a relatively homogenous class based on molecular and electrophysiological properties. They display short-duration action potentials, pronounced adaptation and weak inwardly rectifying current after hyperpolarizing pulses. SOM-expressing interneurons in the hilus are considered to correspond to the hilar perforant path-associated (HIPP) cells [18,21,40,99], and the overall features of SOM-NPY neurons identified in our study match well with the electrophysiological [19,20,100,101] and molecular [14,18,99,101,102] characteristics of HIPP cells. At the functional level, HIPP cells play an essential role in the negative feedback to granule cells [36,94]. A recent study also described a novel type of SOM expressing interneurons projecting in the hilus [16], suggesting that the cluster of NPY-SOM interneurons we identified, could also include HIL cells. These cells exert a strong peri-somatic inhibition onto local GABAergic inhibitory cells [16].

Finally, the most prominent cluster of GABAergic neurons identified by our unsupervised analysis (cluster 3, Figs 2 and 3) gathers neurons expressing NPY, PV, NOS1 and CCK. These neurons exhibit heterogenous firing patterns and our analysis does not reveal any preferential association between molecular markers, suggesting that their expression was independent. PV is strongly associated with basket-cell projection type [17,34,37,39,40,103]. Some studies have also linked hilar commissural associated path (HICAP) interneurons to expression of CCK [40,77] or NOS1 [18,39]. Thus, it seems likely that our third cluster encompasses several classes of interneurons associated with PV, NOS1 and CCK expression, respectively. Surprisingly, however, our data suggest that these types of neurons do not display strongly discriminating electrophysiological properties [20].

These results contrast with the pattern of diversification of GABAergic neurons existing in the rest of the telencephalon, where PV is strongly associated with a very typical fast spiking electrophysiological type [26–28,30,31,53]. Thus, our finding suggest that hilar PV basket cells

do not exhibit a strong fast-spiking characteristic. Indeed, the projection of hilar GABAergic interneurons onto the principal components defining cortical interneurons indicates that PV neurons do not have a clear homolog in the classical cortical classes. These results incite caution in assuming that interneurons expressing a specific marker in the hilus are comparable to classes defined by expression of the same markers in other telencephalic regions. Accordingly, recent studies show that the functional properties of PV basket cells are different in the hilus than in the rest of the forebrain [17,104].

## Potential limitation and conclusions

In the hilus of the dentate gyrus, the search for descriptive markers allowing a clear discrimination between interneuron types is still debated [20,105,106] unlike in other hippocampal areas [18,29].

Here we find that electrophysiological and molecular properties fail to yield a convincing partition of hilar interneurons, while this methodology divides a control sample of cortical neurons into well validated classes. This indicates that hilar GABAergic cells share only limited homology with cortical interneurons.

However, our study does not invalidate the use of classical markers to distinguish interneuron types in the hilus. Indeed, markers such as PV and SOM have been linked to well defined projections pattern in the granule cell layer and molecular layers [16,34–37,39,40]. Moreover, in the present study, the scRT-PCR procedure limited the quality of histological staining and axonal arborizations could not be recovered. It is also important to note that some of the differences that we find between hilar and cortical neurons may result from uncontrolled parameters. Indeed, our cortical and hilar samples were collected under slightly different conditions and respectively in juveniles (P14-P17) and young adults (2 to 3 months). Nevertheless, it is well established that the GABAergic neuronal classes identified in our cortical samples remain stable at later developmental stages [23,24]. Thus, our results indicate that, contrasting with other telencephalic interneurons, hilar GABAergic neurons defined by PV, NOS1 and CCK expression dot not present strongly differentiated electrophysiological properties.

This raises puzzling questions about how GABAergic interneurons differentiate. One possibility is that electrophysiological differentiation between interneuron types is not functionally beneficial in the hilus. Another is that there is some degree of differentiation which the markers used in this study were unable to identify. Recent advances in single-cell RNA sequencing (scRNA-seq) are evolving the concept of neuronal classification [107,108]. Such new tools should promote the identification of new molecular markers suitable to account for the diversity of hilar interneurons and to understand their role in the physiology of the dentate gyrus.

## Supporting information

**S1 Fig. Quantification of GAD67 and NeuN expression across the hilus and granular cell layer of the dentate gyrus. A1-A4:** Representative photomicrograph of GFP and NeuN expression in rostral sections of the dentate gyrus of GAD67-GFP knock-in mice. Each of the following layers was divided into 8 bins using a semiautomated procedure (Materials and methods): **A1:** granular cell layer outer blade (bin 0 to 7), **A2:** hilar outer half (bin 8 to 15), **A3:** hilar inner half (bin 16 to 23) and **A4:** granular cell layer inner blade (bin 24 to 31). The delineated area in A1 is enlarged in **A5**. **B-E:** Histograms of the densities of GFP (grey) and NeuN (black) expressing cells in **A:** rostral, **B:** median and **C:** caudal slices and in **E:** all analyzed areas, for **1:** bin 0 to 7, **2:** bin 8 to 15, **3:** bin 16 to 23 and **4:** bin 24 to 31. **F:** Expression of GAD in NeuN expressing cells. (n = 7 mice, error bars: sem, gcl: granular cell layer, h: hilus, ml: molecular layer).
(TIF)

**S2 Fig. Quantification of GAD67 and NeuN expression across the hilus and granular cell layer of the dentate gyrus. A1-A3:** CR immunostaining in the dentate gyrus of a GAD67 GFP knock-in mouse. In A3, the delineated area is enlarged in the inset. Examples of GAD expressing neurons labeled with CR are pointed out with arrows. **A4:** Densities of GFP expressing cells (grey), CR expressing cells (black outline) and CR positive GFP expressing cells (black) in rostral, median and caudal slices. CR positive cells were not counted in bin 0 to 7 and 24 to 31. The histogram in the bottom right represents the percentage of expression of CR in GAD67 expressing cells in the hilus. **B1-B3:** PV immunostaining in the dentate gyrus of a GAD67-GFP knock-in mouse, the delineated area is enlarged in the inset in B3. Examples of GAD-expressing neurons labeled with PV are pointed out with arrows. **B4:** Histograms representing the densities of GFP (grey) and PV (black) expressing cells in bin 0 to 31, in rostral, median and caudal slices. The histogram at the bottom right represents the percentage of expression of PV in GAD expressing cells in the hilus (black) and granular cell layer (white). **C1-C4.** As described in B1- B4 for SOM. (n = 7 mice/marker, error bars: sem).
(TIF)

**S3 Fig. Quantification of NOS-1 and NPY immunoreactivity in GABAergic neurons along the antero-posterior axis of the dentate gyrus. A1-A3:** NOS-1 immunostaining in the dentate gyrus of a GAD67-GFP knock-in mouse. In A3, the delineated area is enlarged in the inset. Examples of GAD67 expressing neurons labelled with NOS1 are pointed out with arrows. **A4:** Histograms represent the densities of GFP (grey) and NOS1 (black) expressing cells in bin 0 to 31, in rostral, median and caudal slices. The histogram at the bottom right represents the percentage of expression of NOS1 in GAD expressing cells in the hilus (black) and granular cell layer (white). **B1-B4:** As described in A1-A4 for NPY. (n = 7 mice/marker, error bars: sem).
(TIF)

**S4 Fig. Segregation of 3 classes of hilar GABAergic neurons maximizes the distance step in the hierarchical clustering.** Top: Dendrogram of aggregation applying Ward's unsupervised clustering to 51 hilar GABAergic neurons. 16 electrophysiological and 8 molecular parameters were used. The x axis represents individual cells and the y axis the distance of aggregation. **Bottom:** Distance to the closest downstream node after the first 9 nodes of the dendrogram. Distance is maximal after node 2 segregating 3 clusters.
(TIF)

**S5 Fig. Electrophysiological properties of hilar GABAergic neurons.** Values are represented as mean +/- sem. for each parameter and each cluster (yellow: cluster 1; green: cluster 2; purple: cluster 3; RMP: Resting membrane potential; Rm: Input resistance; Tic: Time constant of membrane capacitance; Cm: Membrane capacitance; Gsag: Rectification of hyperpolarization; 1$^{st}$ spike delay: delay to first spike from the onset of current injection at rheobase; Asat: Amplitude of adaptation near saturation; tsat: Time constant of adaptation near saturation; msat: Slope of adaptation near saturation; Csat: intersect of adaptation near saturation; A1: Amplitude of the first spike; D1: Duration of the first spike; AHP1max: Amplitude of after hyperpolarization potential; tAHP1max: Time of maximum after hyperpolarization potential; Amp. Red.: Amplitude reduction between the first and second spike at rheobase; Dur. Inc.: Duration increase between the first and second spike at rheobase; Material and methods).
(TIF)

**S6 Fig. Cluster 1 neurons segregate together with glutamatergic cells in a general classification of hilar neurons. A.** Ward's unsupervised clustering applied to 147 hilar neurons based on their electrophysiological and molecular properties characterized with single cell RT-PCR.

The same 16 electrophysiological and 8 molecular parameters were used as previously (Fig 2). The analysis disclosed 2 main branches. Colored circles represent GABAergic neurons clustered as in Fig 2. (yellow: cluster 1; green: cluster 2; purple: cluster 3). Most cluster 1 neurons are grouped in Branch 1, whereas all neurons from clusters 2 and 3 are assigned to branch 2. **B**. Histogram showing the expression of molecular markers in branch 1 (left; grey) and branch 2 neurons (right; purple). Branch 1 mostly comprises glutamatergic neurons.
(TIF)

**S7 Fig. Electrophysiological properties of cortical GABAergic interneurons.** Values are represented as mean +/- sem. for each parameter and cluster (yellow: cluster 1; green: cluster 2; purple: cluster 3; RMP: Resting membrane potential; Rm: Input resistance; Tic: Time constant of membrane capacitance; Cm: Membrane capacitance; Gsag: Rectification of hyperpolarization; 1$^{st}$ spike delay: delay to first spike from the onset of current injection at rheobase; Asat: Amplitude of adaptation near saturation; tsat: Time constant of adaptation near saturation; msat: Slope of adaptation near saturation; Csat: intersect of adaptation near saturation; A1: Amplitude of the first spike; D1: Duration of the first spike; AHP1max: Amplitude of after hyperpolarization potential; tAHP1max: Time of maximum of after hyperpolarization potential; Amp. Red.: Amplitude reduction between the first and second spike at rheobase; Dur. Inc.: Duration increase between the first and second spike at rheobase; Material and methods).
(TIF)

**S8 Fig. Parameter-wise statistical differences between cortical and hilar GABAergic neurons.** P-values of the statistical test of the difference between hilar and cortical GABAergic interneurons (Mann-Whitney rank-sum test). Values are sorted in ascending order. Different sets of cortical and hilar neurons were used based on markers expression. Marker used were **A.** GAD (i.e. full samples), **B.** PV, **C.** SOM and **D.** NPY. RMP and Csat (red) showed highly significant differences indicating a potential sample wise bias.
(TIF)

**S9 Fig. Removing RMP and Csat has a limited impact on the principal components and centroid-centroid axes.** Clusters of cortical (left) and hilar GABAergic neurons (right) projected on the two first principal components of the cortical population using the full parameter set (top) or a parameter set excluding RMP and Csat (bottom), which might be subjected to population wise biases. Percentage in the x and y axes labels indicate each PC's explained variance Crosses represent population centroids (black: cortex, blue: hilus) and dashed lines represent centroid-centroid axes. Removing RMP and Csat minimally affects the spread of cortical and hilar neurons and the positions of centroids along the 2 first principal components.
(TIF)

**S1 Table. Statistical association between markers in the cluster 3 of hilar GABAergic interneurons.** Association was tested directionally using binomial test (Material and methods). No significant association was detected.
(DOCX)

## Acknowledgments

We thank Yuchio Yanagawa, Cecile Lebrand and Jean-Pierre Hornung for providing the GAD67:GFP animals. The authors are grateful to Bruno Cauli, Fares Sayegh for their comments and discussions on our manuscript. An early version of this manuscript was previously published in the PhD thesis of Clémence Leclerc [109] (https://tel.archives-ouvertes.fr/tel-00833326/document).

## Author Contributions

**Conceptualization:** Quentin Perrenoud, Tania Vitalis, Claire Rampon, Thierry Gallopin.

**Data curation:** Quentin Perrenoud, Clémence Leclerc, Hélène Geoffroy, Tania Vitalis, Kevin Richetin.

**Formal analysis:** Quentin Perrenoud, Clémence Leclerc.

**Funding acquisition:** Claire Rampon, Thierry Gallopin.

**Investigation:** Quentin Perrenoud, Clémence Leclerc, Tania Vitalis, Claire Rampon, Thierry Gallopin.

**Methodology:** Quentin Perrenoud, Clémence Leclerc, Hélène Geoffroy, Tania Vitalis, Thierry Gallopin.

**Project administration:** Claire Rampon, Thierry Gallopin.

**Resources:** Quentin Perrenoud, Claire Rampon, Thierry Gallopin.

**Software:** Quentin Perrenoud.

**Supervision:** Claire Rampon, Thierry Gallopin.

**Validation:** Quentin Perrenoud, Kevin Richetin, Claire Rampon, Thierry Gallopin.

**Visualization:** Quentin Perrenoud, Clémence Leclerc, Hélène Geoffroy, Tania Vitalis, Kevin Richetin, Claire Rampon, Thierry Gallopin.

**Writing – original draft:** Quentin Perrenoud, Clémence Leclerc, Thierry Gallopin.

**Writing – review & editing:** Quentin Perrenoud, Claire Rampon, Thierry Gallopin.

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
