## [Decision Letter · Decision Letter 0]

21 Mar 2022

PONE-D-22-04089Dear Editor(s),

Molecular and electrophysiological features of GABAergic neurons in the dentate gyrus reveal limited homology with cortical interneuronsPLOS ONE

Dear Dr. Gallopin,

Thank you for submitting your manuscript to PLOS ONE. After careful consideration, we feel that it has merit but does not fully meet PLOS ONE’s publication criteria as it currently stands. Therefore, we invite you to submit a revised version of the manuscript that addresses the points raised during the review process.

We look forward to receiving your revised manuscript.

Kind regards,

Giuseppe Biagini, MD

Academic Editor

PLOS ONE

Journal Requirements:

- https://tel.archives-ouvertes.fr/tel-00833326/document

In your revision ensure you cite all your sources (including your own works), and quote or rephrase any duplicated text outside the methods section. Further consideration is dependent on these concerns being addressed.

"This work was supported by Association France Alzheimer (FA2010 to C.R. and T.G.), the Centre National de la Recherche Scientifique, Ecole Supérieure de Physique et de Chimie Industrielle and University of Toulouse 3. We thank Yuchio Yanagawa, Cecile Lebrand and Jean-Pierre Hornung for providing the GAD67:GFP animals. The authors are grateful to Bruno Cauli, Fares Sayegh for their comments and discussions on our manuscript."

"This work was supported by a grant from the France Alzheimer Association (www.francealzheimer.org/). The work was also supported by the Centre National de la Recherche Scientifique (www.cnrs.fr/fr), the University of Toulouse 3 (www.univ-toulouse.fr) and the Ecole Supérieure de Physique et Chimie Industrielle-Paris (ESPCI Paris, www.espci.psl.eu/en). The funders had no role in study design, data collection and analysis, decision to publish, or preparation of the manuscript."

Reviewers' comments:

Reviewer's Responses to Questions

**Comments to the Author**

1. Is the manuscript technically sound, and do the data support the conclusions?

Reviewer #1: No

Reviewer #2: Yes

2. Has the statistical analysis been performed appropriately and rigorously? 

Reviewer #1: Yes

Reviewer #2: Yes

3. Have the authors made all data underlying the findings in their manuscript fully available?

Reviewer #1: Yes

Reviewer #2: Yes

4. Is the manuscript presented in an intelligible fashion and written in standard English?

Reviewer #1: Yes

Reviewer #2: Yes

5. Review Comments to the Author

Reviewer #1: The MS by Perrenoud et al reports data on the distribution, and molecular and cellular properties of GABAergic neurons in the hilus of the mouse hippocampus. On the background of several former studies concerned with the classification of inhibitory neurons in the hippocampus, the authors have provided an extended description of such neurons based on multidisciplinary methods like patch clamping plus single cell PCR, and immunohistochemistry.

The present study is based on data analysis with complex methods that, I suspect, are beyond the grasp and interest of the average PLOS reader. Nevertheless, despite all these efforts, we are left with conclusions that appear to increase our doubt rather than clarifying our understanding of GABAergic neurons in this brain area. I just quote a couple of sentences in the Discussion: on p.23: “The other group, which represents the majority of our sample (30 neurons out of 51), shows a variety of electrophysiological profiles and uncorrelated expression of molecular markers.” And on p.24: “It is important to stress however, that based on our data, the distinction between GABAergic and glutamatergic cells may be misleading.” These statements would demonstrate that relying on the expression of certain peptide biomarkers (and other proteins like membrane transporters) to characterize inhibitory interneurons is unreliable. This is a sharp departure from current knowledge. Thus, much more conclusive evidence is required to support the MS statements.

In particular, the authors have not used confocal microscopy to study hippocampal neurons, a situation which limits the resolution of their data. Indeed, the authors indicate that “Counting of NeuN-expressing cells was restricted to the hilus as cell densities were too high in the GCL and individual cells could not be distinguished.”

Furthermore, neuronal projections could not be traced with biocytin staining because this substance was applied for a short time only (10 min) via the patch electrode. What was the usefulness of adding biocytin if it could not be used?

Many papers nowadays contain a paragraph at the end of the discussion to clearly list the study limitations. It is recommended that this addition is included here. Moreover, the abstract should have a conclusive sentence with a few words of caution about the data implications.

While the MS is generally well written, there are a few points the authors should correct: somas, 10m (meters instead of min ?), exemple in Fig 3 E2.

It is unclear why electrophysiological values were not corrected for junction potential? P. 6 (middle).

The text formatting is not very helpful as it contains the Fig legends embedded in the results while the actual Figs are at the end of the MS.

Reviewer #2: The paper addresses the question of whether- and how- it is possible to classify hilar GABAergic neurons based on electrophysiological data and molecular markers expression. The analysis is further corroborated by comparison with the application of the same analysis to a dataset coming from cortical cells. The analysis, based on unsupervised clustering and resampling methods, adds new information to the existing field and contributes to the understanding of GABAergic hilar interneurons.

Main Comments:

Authors should add the percentage of variance explained by PC1 and PC2 to each graph. If this is low, it might impact the strength of the study. This point should be commented.

Authors state that there are slight differences in the recording conditions between hilar and cortical neurons. Age difference (P14-P17 vs 2-3 months) can actually be quite an important factor. It should be discussed how this impacts the conclusions on hilar cells.

Minor Comments:

“>>>” indicates how the word/phrase should be changed

Pg 6:

Line 11: Slice >>> Slices (were maintained….)

10m >>> 10 min, according to the International System of Units

Pg 8:

Numbers 1-16 of the electrophysiological parameters are not consistently put in the same position relative to the measure they refer to. It might be clearer to put the number after the measure in all cases [for ex: Sag Ratio (5) was computed…… Rheobase (6) was defined…. Time constant of late adaptation (8) etc]

Spike durations� spike duration (was computed)

Pg 9:

Marker >>> markers (We concluded that the correlation structure between marker was significant)

“the angle between of the principal component capturing the most variance (PC1) in each sample”: it is not clear "between" what.

“We concluded in a significant difference in PC1 angle and centroid distances when”: this sentence is not clearly written in English and should be reworded.

Pg 10:

“when their actual value was” >>> when their value was.

Pg 15

A variety of firing pattern >>> A variety of firing patterns

the expression of all pair of >>> the expression of all pairs of

Pg 17:

hilar GABAergic neurons does not related >>> hilar GABAergic neurons does not relate

Pg 19:

We found that the angle between… >>> We found that the cosine of the angle between…

This value was very significantly different from H0 >>> This value was very significantly different from the expected cosine value in H0 (or similar)

Pg 22:

“Such studies have convincingly shown that PV- and SOM-expressing interneurons target the same principal neuron sub-compartments than interneurons expressing these markers in the rest of the telencephalon” >>> Such studies have convincingly shown that PV- and SOM-expressing interneurons target the same principal neuron sub-compartments IN THE HILLUS than interneurons expressing these markers in the rest of the telencephalon

Fig. 3E2 Exemple >>> Example

Graphs might be more clear if significance stars are horizontal (***), as is standard, rather than vertical.

6. PLOS authors have the option to publish the peer review history of their article (what does this mean?). If published, this will include your full peer review and any attached files.

Reviewer #1: No

Reviewer #2: No

---

## [Author Response · Author response to Decision Letter 0]

19 Apr 2022

Dear Dr Biagini

Thank you very much for considering our manuscript for publication in PLOS ONE. Following the instructions given in your decision letter, you’ll find below our point-by-point response to the concerns raised by yourself as well by the reviewers. We have fully addressed all issues of the reviewers and hope that the manuscript is now acceptable for publication.

In order to make the tracking of corrections as clear as possible, we have highlighted in the revised manuscript in yellow the text that has been added, and crossed out the text that has been removed and highlighted in green. As a result, the page numbers that are referenced in the letter take into account this follow-up of the corrections

Requirements:

*Query 1: Please ensure that your manuscript meets PLOS ONE's style requirements, including those for file naming. The PLOS ONE style templates can be found at

-Authors' responses:

The manuscript conforms to the PLOS ONE’s style requirements to the best of our efforts. Please make us aware of any unconformities that might remain.

*Query 2: We noticed you have some minor occurrence of overlapping text with the following previous publication(s), which needs to be addressed: - https://tel.archives-ouvertes.fr/tel-00833326/document

In your revision ensure you cite all your sources (including your own works), and quote or rephrase any duplicated text outside the methods section. Further consideration is dependent on these concerns being addressed.

-Authors' responses:

We apologize for this oversight. The publication mentioned is the PhD thesis of Clémence Leclerc who is the second author in our study. This PhD thesis contains an early version of the present manuscript which explains the overlap. In the revised version of our manuscript, we have included an explicit reference to this previous version in the acknowledgments (p36, L5-7) and in the References (p40, L34-36)

An analysis performed on the copyleaks website reveals that, apart from the method section, our manuscript in its present state contains only 10.3% overlapping text with this earlier version.

*Query 3: Thank you for stating the following in the Acknowledgments Section of your manuscript: "This work was supported by Association France Alzheimer (FA2010 to C.R. and T.G.), the Centre National de la Recherche Scientifique, Ecole Supérieure de Physique et de Chimie Industrielle and University of Toulouse 3. We thank Yuchio Yanagawa, Cecile Lebrand and Jean-Pierre Hornung for providing the GAD67:GFP animals. The authors are grateful to Bruno Cauli, Fares Sayegh for their comments and discussions on our manuscript." We note that you have provided funding information that is not currently declared in your Funding Statement. However, funding information should not appear in the Acknowledgments section or other areas of your manuscript. We will only publish funding information present in the Funding Statement section of the online submission form. Please remove any funding-related text from the manuscript and let us know how you would like to update your Funding Statement. Currently, your Funding Statement reads as follows:

"This work was supported by a grant from the France Alzheimer Association (www.francealzheimer.org/). The work was also supported by the Centre National de la Recherche Scientifique (www.cnrs.fr/fr), the University of Toulouse 3 (www.univ-toulouse.fr) and the Ecole Supérieure de Physique et Chimie Industrielle-Paris (ESPCI Paris, www.espci.psl.eu/en). The funders had no role in study design, data collection and analysis, decision to publish, or preparation of the manuscript." 

-Authors' responses:

Thank you very much for pointing this error. The funding statement has now been removed from the acknowledgment section of the manuscript. As also mentioned in the cover letter, we do not wish to amend our funding statement. 

*Query 4: Please amend either the title on the online submission form (via Edit Submission) or the title in the manuscript so that they are identical. 

-Authors' responses:

As far as we could verify it, our study’s title in the manuscript and in our copy of the submission form are identical. In case there is any inconsistency that we might have skipped over, the title of the manuscript should be: “Molecular and electrophysiological features of GABAergic neurons in the dentate gyrus reveal limited homology with cortical interneurons”

*Query 5: We note that you have included the phrase “data not shown” in your manuscript. Unfortunately, this does not meet our data sharing requirements. PLOS does not permit references to inaccessible data. We require that authors provide all relevant data within the paper, Supporting Information files, or in an acceptable, public repository. Please add a citation to support this phrase or upload the data that corresponds with these findings to a stable repository (such as Figshare or Dryad) and provide and URLs, DOIs, or accession numbers that may be used to access these data. Or, if the data are not a core part of the research being presented in your study, we ask that you remove the phrase that refers to these data.

-Authors' responses:

A table detailing the result of the data in question was added to the online data of the manuscript ("CellCounts_Histology.xlsx")

(https://github.com/QPerrenoud/Perrenoud_Hilus_DataCode/blob/master/CellCounts_Histology.xlsx). The mention ‘data not shown’ has now been replaced by ‘online data’. (p12, L18).

Reviewer #1: 

*Query 1: The MS by Perrenoud et al reports data on the distribution, and molecular and cellular properties of GABAergic neurons in the hilus of the mouse hippocampus. On the background of several former studies concerned with the classification of inhibitory neurons in the hippocampus, the authors have provided an extended description of such neurons based on multidisciplinary methods like patch clamping plus single cell PCR, and immunohistochemistry.

The present study is based on data analysis with complex methods that, I suspect, are beyond the grasp and interest of the average PLOS reader. Nevertheless, despite all these efforts, we are left with conclusions that appear to increase our doubt rather than clarifying our understanding of GABAergic neurons in this brain area. I just quote a couple of sentences in the Discussion: on p.23: “The other group, which represents the majority of our sample (30 neurons out of 51), shows a variety of electrophysiological profiles and uncorrelated expression of molecular markers.” And on p.24: “It is important to stress however, that based on our data, the distinction between GABAergic and glutamatergic cells may be misleading.” These statements would demonstrate that relying on the expression of certain peptide biomarkers (and other proteins like membrane transporters) to characterize inhibitory interneurons is unreliable. This is a sharp departure from current knowledge. Thus, much more conclusive evidence is required to support the MS statements.

-Authors' responses:

We thank the reviewer for his/her criticism and suggestions and provide a point-by-point answer below. We regret to read that reviewer 1 is left with what seems to be a misunderstanding of our conclusions. We are fully aware that our study raises more questions than it clarifies our understanding of how hilar GABAergic neurons diversify, and that is one of the reasons we consider this data valuable. As clearly stated in our abstract, the primary conclusion of our article is that using unsupervised clustering based on electrophysiological and molecular properties “fails to provide a meaningful partition of hilar interneurons”. This finding was unexpected to us as this methodology has widely been applied to classify GABAergic interneurons in the cortex and in other parts of the hippocampus. This sorting approach was applied in our study to interneurons from the barrel cortex, leading to a well-validated classification. This supports our second major conclusion that the failure to segregate hilar GABAergic interneurons stems from their unique properties, rather than from our methodology. It should be noted that, contrary to what the reviewer seems to point out, and as clearly stated in our Introduction and Discussion, and further clarified in the Abstract (p2, L10-16) and Conclusion (p35, L6-14) of the revised manuscript, we do not consider our study to invalidate the use of peptides and calcium binding proteins expressions for the classification of hilar interneurons as some of them (PV, SOM) have been shown to correlate well with axonal projection targets. Our study does however strongly suggest that these markers do not correlate with robustly discriminating electrophysiological properties in the hilus, contrasting with other brain regions such as the neocortex. While our conclusions are mainly negative, we believe that they are supported by our data and that they provide valuable information to guide the design of future studies. We believe that PLOS ONE is a journal fully suitable for the publication of our data as the importance of the publication of negative results is clearly stated in PLOS ONE editorial policies (https://journals.plos.org/plosone/s/criteria-for-publication). 

*Query 2: In particular, the authors have not used confocal microscopy to study hippocampal neurons, a situation which limits the resolution of their data. Indeed, the authors indicate that “Counting of NeuN-expressing cells was restricted to the hilus as cell densities were too high in the GCL and individual cells could not be distinguished.”

-Authors' responses:

We agree with the reviewer that the lack of confocal microscopy limits the resolution of our histological data in the granule cell layer. Nonetheless, it is important to point out that this limitation does not invalidate our main conclusions which are based primarily on patch-clamp and single cell RT-PCR data, rather than on histological analysis. In addition, the focus of our study is primarily on interneurons located in the hilus rather than in the granule cell layer.

*Query 3: Furthermore, neuronal projections could not be traced with biocytin staining because this substance was applied for a short time only (10 min) via the patch electrode. What was the usefulness of adding biocytin if it could not be used?

-Authors' responses:

The absence of neuronal tracing is indeed a limitation of our study. While this was already mentioned in the discussion, we followed the suggestion of the reviewer, and clearly stated this again in a paragraph at the end of the Discussion, dedicated to the limitations of the study (p35, L3-32). As explained there, single cell RT-PCR makes it difficult to obtain acceptable staining of the axonal arbor (p35, L15-16). Biocytin was added to the recording solution because we intended to perform a multiparametric analysis, including analysis of the axonal arborization of the cells, which is known to identify certain types of interneurons in the hilus. Unfortunately, the low tracer diffusion inherent in cytoplasm aspiration and the short recording period required to ensure the viability of the mRNA prevented achieving usable reconstructions. 

*Query 4: Many papers nowadays contain a paragraph at the end of the discussion to clearly list the study limitations. It is recommended that this addition is included here. Moreover, the abstract should have a conclusive sentence with a few words of caution about the data implications.

-Authors' responses:

According to the suggestion of the reviewer, we have modified the conclusion of the Discussion to include a statement of the potential limitations of our study, including the points raised by the reviewer (p32, L3-32). We have also made it clear in the Abstract that our study does not invalidate the use of molecular markers to classify hilar neurons (p2, L10-16).

*Query 5: While the MS is generally well written, there are a few points the authors should correct: somas, 10m (meters instead of min ?), exemple in Fig 3 E2.

-Authors' responses:

We thank the reviewer for pointing out these mistakes. The manuscript was corrected accordingly.

*Query 6: It is unclear why electrophysiological values were not corrected for junction potential? P. 6 (middle). 

-Authors' responses:

We thank the reviewer for pointing this out. Correcting for junction potentials is possible and indicated when one has full control of the extracellular and intracellular milieu. Inside out and outside out patch clamp configurations clearly meet this condition. However, this condition is not achieved in the whole cell configuration since the junction potential is also affected by ionized compounds within the cytoplasm. 

Nevertheless, we now clearly state in the methods the magnitude of the correction that can be inferred from the content of the internal and external solutions (p6, L27-29). The matlab script we used for this calculation is now included in the online data. A correction for junction potential would bias the values of the resting membrane potential (RMP) of hilar and cortical neurons by -14.8mV and -15mV respectively. 

As stated in the methods (p6, L29-30), this slight difference (0.2mV) does not impact our analyses since RMP was not considered for the comparison of hilar and cortical neurons. Besides, as the mean RMP of hilar and cortical neurons were -51.4mV and -66.9mV respectively, correcting for junction potential would further amplify this difference and strengthen our conclusion that hilar and cortical interneurons have different electrophysiological properties.

*Query 7: The text formatting is not very helpful as it contains the Fig legends embedded in the results while the actual Figs are at the end of the MS.

-Authors' responses:

We apologize for this inconvenience, but unless we are mistaken, this configuration was requested by the journal.

Reviewer #2: 

*Query 1: The paper addresses the question of whether- and how- it is possible to classify hilar GABAergic neurons based on electrophysiological data and molecular markers expression. The analysis is further corroborated by comparison with the application of the same analysis to a dataset coming from cortical cells. The analysis, based on unsupervised clustering and resampling methods, adds new information to the existing field and contributes to the understanding of GABAergic hilar interneurons.

-Authors' responses:

We are grateful for this positive appreciation of our study and thank the reviewer for his/her comments and suggestions. Our point-by-point answer is detailed below. 

Main Comments: 

*Query 2: Authors should add the percentage of variance explained by PC1 and PC2 to each graph. If this is low, it might impact the strength of the study. This point should be commented. 

-Authors' responses:

We thank the reviewer for this excellent suggestion. The percentage of variance of each PC has now been added to figures 4, 5 and S9. As can be seen there, PC1 and PC2 collectively represent about a third of the variance in each case examined (Fig 4: Full Sample: 32.8%, Fig 5: CCK: 38.6%, NOS1: 36.9%, PV: 30%, SOM: 30.4%, NPY: 31%). 

We would like to point out that, PCA plots in these figures were used for illustration only. Those plots only use principal components derived from our cortical sample and do not directly relate to the statistical analysis illustrated in Fig 4D2. This analysis is based on the cosine angle between the first principal components of our cortical AND hilar samples. 

To clarify this further, we now state the magnitude of PC1 in our cortical and hilar samples in the result (p25, L27-32). As described there PC1 accounts respectively for 22.1% and 23.8% of the variance of our cortical and hilar sample. This is substantial, but leaves aside a large fraction of the variance. We feel that this provides the rationale for the introduction of a second test based on Mahalanobis distances and described in figure 5.

*Query 3: Authors state that there are slight differences in the recording conditions between hilar and cortical neurons. Age difference (P14-P17 vs 2-3 months) can actually be quite an important factor. It should be discussed how this impacts the conclusions on hilar cells.

-Authors' responses:

We fully agree with the reviewer that age difference might be a confounding factor in our comparison. Following the suggestions of reviewer #1, the end of our discussion now provides a paragraph listing this and other issues as potential limitations to our study (p35, L6-32). According to the discussion therein, it is well established that the classes that we identify in the juvenile cortical sample remain stable at later developmental stages. This suggest that while animal’s age might indeed affect the magnitude of the differences observed between hilar and cortical interneurons, the main conclusions of our study would not be affected.

*Query 4: Minor Comments:

“>>>” indicates how the word/phrase should be changed

-Authors' responses:

We warmly thank the reviewer for pointing out these mistakes. The revised manuscript was changed accordingly

*Pg 6:

Line 11: Slice >>> Slices (were maintained….)

-Authors' responses:

Corrected (p6, L11)

*10m >>> 10 min, according to the International System of Units

-Authors' responses:

Corrected (p6, L32)

*Pg 8:

Numbers 1-16 of the electrophysiological parameters are not consistently put in the same position relative to the measure they refer to. It might be clearer to put the number after the measure in all cases [for ex: Sag Ratio (5) was computed…… Rheobase (6) was defined…. Time constant of late adaptation (8) etc

Spike durations� spike duration (was computed)

-Authors' responses:

Corrected (p8)

*Pg 9:

Marker >>> markers (We concluded that the correlation structure between marker was significant)

-Authors' responses:

Corrected (p9, L26)

*“the angle between of the principal component capturing the most variance (PC1) in each sample”: it is not clear "between" what.

-Authors' responses:

This sentence was changed to “the angle between the first principal components (PC1) of each of the two samples” (p9, L32-34)

*“We concluded in a significant difference in PC1 angle and centroid distances when”: this sentence is not clearly written in English and should be reworded.

-Authors' responses:

This sentence was changed to “Values were considered significant when superior to the 95th percentile of the resampled distribution.” (p10, L4-7)

*Pg 10: 

“when their actual value was” >>> when their value was.

-Authors' responses:

Corrected (p10, L6)

*Pg 15: 

A variety of firing pattern >>> A variety of firing patterns 

-Authors' responses:

corrected (p17, L20)

*the expression of all pair of >>> the expression of all pairs of

-Authors' responses:

corrected (p17, L25)

*Pg 17:

hilar GABAergic neurons does not related >>> hilar GABAergic neurons does not relate

-Authors' responses:

corrected (p21, L10)

*Pg 19:

We found that the angle between… >>> We found that the cosine of the angle between…

This value was very significantly different from H0 >>> This value was very significantly different from the expected cosine value in H0 (or similar)

-Authors' responses:

Corrected (p25, L21-22)

*Pg 22: 

“Such studies have convincingly shown that PV- and SOM-expressing interneurons target the same principal neuron sub-compartments than interneurons expressing these markers in the rest of the telencephalon” >>> Such studies have convincingly shown that PV- and SOM-expressing interneurons target the same principal neuron sub-compartments IN THE HILLUS than interneurons expressing these markers in the rest of the telencephalon

-Authors' responses:

Corrected (P31 L6)

*Fig. 3E2 Exemple >>> Example

-Authors' responses:

Corrected (p19-20)

*Graphs might be more clear if significance stars are horizontal (***), as is standard, rather than vertical.

-Authors' responses:

Corrected 

We thank the reviewer for pointing this out. The figures were modified accordingly.

---

## [Decision Letter · Decision Letter 1]

5 Jun 2022

PONE-D-22-04089R1Molecular and electrophysiological features of GABAergic neurons in the dentate gyrus reveal limited homology with cortical interneuronsPLOS ONE

Dear Dr. Gallopin,

Thank you for submitting your manuscript to PLOS ONE. After careful consideration, we feel that it has merit but does not fully meet PLOS ONE’s publication criteria as it currently stands. Therefore, we invite you to submit a revised version of the manuscript that addresses the points raised during the review process.

We look forward to receiving your revised manuscript.

Kind regards,

Giuseppe Biagini, MD

Academic Editor

PLOS ONE

Journal Requirements:

Reviewers' comments:

Reviewer's Responses to Questions

**Comments to the Author**

1. If the authors have adequately addressed your comments raised in a previous round of review and you feel that this manuscript is now acceptable for publication, you may indicate that here to bypass the “Comments to the Author” section, enter your conflict of interest statement in the “Confidential to Editor” section, and submit your "Accept" recommendation.

Reviewer #1: All comments have been addressed

Reviewer #2: (No Response)

2. Is the manuscript technically sound, and do the data support the conclusions?

Reviewer #1: (No Response)

Reviewer #2: Yes

3. Has the statistical analysis been performed appropriately and rigorously? 

Reviewer #1: (No Response)

Reviewer #2: Yes

4. Have the authors made all data underlying the findings in their manuscript fully available?

Reviewer #1: (No Response)

Reviewer #2: Yes

5. Is the manuscript presented in an intelligible fashion and written in standard English?

Reviewer #1: (No Response)

Reviewer #2: Yes

6. Review Comments to the Author

Reviewer #1: (No Response)

Reviewer #2: The authors have addressed the points raised. I still have a few minor requests.

1) The percentage of variance explained by PC1 and PC2 was not added to the graph in Fig. 2D. Unless there is a specific reason for this, it would be nice to have them.

2) I appreciated the clarification that the PCA plots (in Fig. 4C, for example) are just a graphical representation and do not relate to a statistical analysis. To help the reader navigate this analysis, the figures/text could better represent this point. I suggest that the authors write the % explained by the PC only in the graph relating to the cortex (so, for ex. in Fig 4C, “PC Cortex (22.1%)” would have to be moved to the graph above). The graphs below would not have the % since it does not relate to the representation in the graph. If my understanding is correct, the analysis was done in two steps. First, the PCs (and the % variance explained) for cortical neurons were determined. Second, hilar neurons were projected on the parameter space of the Cortex PCs to visually inspect the superimposition of the two sets of dots. This clarification, although a little pedantic, would help a non-expert reader understand better.

7. PLOS authors have the option to publish the peer review history of their article (what does this mean?). If published, this will include your full peer review and any attached files.

Reviewer #1: No

Reviewer #2: No

---

## [Author Response · Author response to Decision Letter 1]

16 Jun 2022

Thank you very much for your renewed support of the publication of our manuscript in PLOS ONE. Please find here, our response to the additional suggestions raised by reviewer #2.

In order to make the tracking of corrections as clear as possible, we have highlighted in the revised manuscript in yellow the text that has been added, and crossed out the text that has been removed and highlighted in green. As a result, the page numbers that are referenced below take into account this follow-up of the corrections

Reviewer #2: 

The authors have addressed the points raised. I still have a few minor requests.

1) The percentage of variance explained by PC1 and PC2 was not added to the graph in Fig. 2D. Unless there is a specific reason for this, it would be nice to have them.

We are grateful to the reviewer to point this out. The percentage of the variance of our hilar interneuron sample explained by PC1 and PC2 has been added to the graphs in Figure 2D. As can be seen there, and further clarified in the results (P14, L5-6) PC1 and PC2 accounted for 19.2% and 18% of the variance, respectively. 

2) I appreciated the clarification that the PCA plots (in Fig. 4C, for example) are just a graphical representation and do not relate to a statistical analysis. To help the reader navigate this analysis, the figures/text could better represent this point. I suggest that the authors write the % explained by the PC only in the graph relating to the cortex (so, for ex. in Fig 4C, “PC Cortex (22.1%)” would have to be moved to the graph above). The graphs below would not have the % since it does not relate to the representation in the graph. If my understanding is correct, the analysis was done in two steps. First, the PCs (and the % variance explained) for cortical neurons were determined. Second, hilar neurons were projected on the parameter space of the Cortex PCs to visually inspect the superimposition of the two sets of dots. This clarification, although a little pedantic, would help a non-expert reader understand better.

We thank the reviewer for this excellent suggestion. Following the reviewers request, the percentage of variance is now only adjoined to the plots of our cortical sample in Figure 4 and 5. In addition, following the reviewer’s insight, we have reformulated our description of our PCA visualization in the results of the revised manuscript to better describe the two steps of the procedure (P18, L12-16). We have also specified in the corresponding legends the nature of the axes of the figures 2D (P15, L2-3), 4C (P18, L35-36) and 5B (P20, L37). 

We feel that this clarifies our analyses and make our manuscript more understandable.

---

## [Decision Letter · Decision Letter 2]

22 Jun 2022

Molecular and electrophysiological features of GABAergic neurons in the dentate gyrus reveal limited homology with cortical interneurons

PONE-D-22-04089R2

Dear Dr. Gallopin,

We’re pleased to inform you that your manuscript has been judged scientifically suitable for publication and will be formally accepted for publication once it meets all outstanding technical requirements.

Kind regards,

Giuseppe Biagini, MD

Academic Editor

PLOS ONE

Additional Editor Comments (optional):

Reviewers' comments:

Reviewer's Responses to Questions

**Comments to the Author**

1. If the authors have adequately addressed your comments raised in a previous round of review and you feel that this manuscript is now acceptable for publication, you may indicate that here to bypass the “Comments to the Author” section, enter your conflict of interest statement in the “Confidential to Editor” section, and submit your "Accept" recommendation.

Reviewer #2: All comments have been addressed

2. Is the manuscript technically sound, and do the data support the conclusions?

Reviewer #2: (No Response)

3. Has the statistical analysis been performed appropriately and rigorously? 

Reviewer #2: (No Response)

4. Have the authors made all data underlying the findings in their manuscript fully available?

Reviewer #2: (No Response)

5. Is the manuscript presented in an intelligible fashion and written in standard English?

Reviewer #2: (No Response)

6. Review Comments to the Author

Reviewer #2: (No Response)

7. PLOS authors have the option to publish the peer review history of their article (what does this mean?). If published, this will include your full peer review and any attached files.

Reviewer #2: No

---

## [Editor Report · Acceptance letter]

1 Jul 2022

PONE-D-22-04089R2 

Molecular and electrophysiological features of GABAergic neurons in the dentate gyrus reveal limited homology with cortical interneurons 

Dear Dr. Gallopin:

I'm pleased to inform you that your manuscript has been deemed suitable for publication in PLOS ONE. Congratulations! Your manuscript is now with our production department. 

Kind regards, 

on behalf of

Dr. Giuseppe Biagini 

Academic Editor

PLOS ONE